# Cholesterol-rich lysosomes induced by respiratory syncytial virus promote viral replication by blocking autophagy flux

Lifeng Chen [1,2,4], Jingjing Zhang [1,4], Weibin Xu [1,4], Jiayi Chen [1], Yujun Tang [1], Si Xiong [1], Yaolan Li [1], Hong Zhang [2] ✉, Manmei Li [1] ✉ & Zhong Liu [1,3] ✉

Respiratory syncytial virus (RSV) hijacks cholesterol or autophagy pathways to facilitate optimal replication. However, our understanding of the associated molecular mechanisms remains limited. Here, we show that RSV infection blocks cholesterol transport from lysosomes to the endoplasmic reticulum by downregulating the activity of lysosomal acid lipase, activates the SREBP2–LDLR axis, and promotes uptake and accumulation of exogenous cholesterol in lysosomes. High cholesterol levels impair the VAP-A-binding activity of ORP1L and promote the recruitment of dynein–dynactin, PLEKHM1, or HOPS VPS39 to Rab7–RILP, thereby facilitating minus-end transport of autophagosomes and autolysosome formation. Acidification inhibition and dysfunction of cholesterol-rich lysosomes impair autophagy flux by inhibiting autolysosome degradation, which promotes the accumulation of RSV fusion protein. RSV-F storage is nearly abolished after cholesterol depletion or knockdown of LDLR. Most importantly, the knockout of LDLR effectively inhibits RSV infection in vivo. These findings elucidate the molecular mechanism of how RSV co-regulates lysosomal cholesterol reprogramming and autophagy and reveal LDLR as a novel target for anti-RSV drug development.

Lysosomes are single membrane-bound organelles characterized by an acidified milieu (pH: 4.5–5.5) and responsible for the degradation of biological macromolecules and maintenance of cellular homeostasis[1–5]. Lysosomes are essential for different types of autophagy, including nonspecific macroautophagy, microautophagy, substrate-specific forms of autophagy (e.g., mitophagy), micropinocytosis, and chaperone-mediated autophagy[6]. As sites for degradation in these autophagic pathways, lysosomes fuse with mature autophagosomes to form autolysosomes and release acid hydrolases to degrade autophagosome contents, such as abnormal proteins and damaged organelles[7,8]. Therefore, lysosome damage or dysfunction is an important cause of impaired autophagy flux[9].

Lysosomes act as key transit points for intracellular cholesterol trafficking[10]. Lysosomal cholesterol levels are determined by an uptake–export balance. Briefly, cholesterol acquisition by lysosomes is mainly achieved via cell surface receptor-mediated uptake of low-density lipoprotein (LDL)[11]. The exogenous cholesterol carried by LDL is hydrolyzed by lysosomal acid lipase (LAL) to release free cholesterol[12–14]. Inside the lumen of the lysosome, Niemann-Pick C2 (NPC2) accepts and transports the LDL cholesterol to NPC1[15–17]; the

[1]State Key Laboratory of Bioactive Molecules and Druggability Assessment & College of Pharmacy, Jinan University, Guangzhou, China. [2]Department of Dermatology, The First Affiliated Hospital, Jinan University, Guangzhou, China. [3]Guangdong Provincial Key Laboratory of Bioengineering Medicine & College of Life Science and Technology, Jinan University, Guangzhou, China. [4]These authors contributed equally: Lifeng Chen, Jingjing Zhang, Weibin Xu.
✉e-mail: tzhangh@jnu.edu.cn; jnulimanmei1209@126.com; tliuzh@jnu.edu.cn

N-terminal domain of NPC1 binds and inserts LDL cholesterol directly into the limiting membrane[18], followed by transport to the endoplasmic reticulum (ER), plasma membrane, and other cellular compartments. Our understanding of cholesterol export from lysosomes is currently incomplete. Oxysterol-binding protein (OSBP), OSBP-related proteins (ORP1L, ORP2, ORP5, and ORP6), Hrs, VPS4, and peroxisomes are known to be involved in cholesterol egress from lysosomes[19–21]. Disruption at any stage of this process results in massive accumulation of cholesterol within the lysosomal lumen or its limiting membrane. On the other hand, low ER cholesterol levels, caused by the blockade of cholesterol export, activate sterol regulatory element-binding protein 2 (SREBP2, a core transcription factor involved in cholesterol metabolism), which promotes the transcription of protein-encoding genes (e.g., *LDLR*) and further exacerbates LDL uptake and lysosomal cholesterol accumulation[22–26]. Recent work has shown that lysosomal cholesterol accumulation prevents autophagy flux by reducing autolysosome fusion[27]. The function of ORP1L in controlling autophagosome positioning, motility, maturation, and fusion with lysosomes depends on the content of cholesterol in lysosomes[28]. Moreover, cholesterol accumulation is closely associated with lysosomal dysfunction[29]. Thus, lysosomal cholesterol metabolism is involved in the regulation of autophagic/lysosomal function in host cells.

Human respiratory syncytial virus (RSV) is an enveloped, negative-sense RNA virus that belongs to the *Pneumovirus* genus of the *Paramyxoviridae* family. It is the most prevalent cause of respiratory illnesses such as bronchitis and pneumonia in infants, older populations, and immunocompromised individuals[30–33]. Despite many efforts to develop drugs against RSV, no therapeutic interventions are available[34,35], probably because the host determinant of RSV infection is not well understood. Therefore, it is necessary to clarify the host cell molecules and signaling pathways that regulate RSV infection. Indeed, a critical role of cholesterol-rich microdomains (lipid rafts) has been described for the entry or release of RSV[36,37], and it was recently found that cholesterol is required for RSV infectivity and stability[38]. Moreover, important advances have been made regarding the role of autophagy in RSV infection[39–41]. However, how lysosomal cholesterol metabolism and autophagy co-regulate RSV infection and the associated molecular mechanisms remain unclear.

Here, we report that the effect of lysosomal cholesterol metabolism on RSV infection is through the regulation of autophagy flux. Our results indicate that RSV infection of host cells blocks cholesterol transport from lysosomes to the ER, activates the SREBP2−LDLR axis, and promotes the uptake and accumulation of exogenous cholesterol in lysosomes. Acidification inhibition and dysfunction of cholesterol-rich lysosomes impair autophagy flux, which creates a site for RSV fusion (F) protein storage. The accumulation of RSV-F in lysosomes is nearly abolished after the knockdown of LDLR. Importantly, the knockout of LDLR effectively inhibits RSV infection in vivo. These findings identify the molecular mechanism by which RSV co-regulates lysosomal cholesterol reprogramming and autophagy and reveal LDLR as a novel target for drug development to treat RSV infection.

## Results

### RSV-induced cholesterol accumulation in lysosomes facilitates the hiding of RSV-F protein in infected cells

To determine whether RSV infection reprograms cholesterol metabolism, filipin III (a fluorescent probe for cholesterol)[42] was used to visualize cholesterol and confirm the distribution of cholesterol in three cell lines: HEp-2 cells, human bronchial epithelial cells (16HBE), and normal primary HBECs. Using fluorescence microscopy, we found that cholesterol was dispersed throughout the cytoplasm of mock-infected cells. In RSV-infected cells, the fluorescence intensity of filipin significantly increased and the probe colocalized with LAMP1 (the marker protein of lysosome), indicative of cholesterol accumulation in lysosomes (Fig. 1a−f). Notably, RSV infection almost completely

prevented the colocalization of filipin and calreticulin (the marker protein of ER) (Supplementary Fig. 1). These results suggest that RSV blocks cholesterol transport from lysosomes to the ER in infected cells. As a positive control for lysosomal cholesterol export, the application of U18666A, an inhibitor of NPC1 (a cholesterol transporter)[43], also led to enhanced cholesterol signals and accumulation of cholesterol in lysosomes. Interestingly, co-staining of cells with antibodies against viral proteins (RSV-G, RSV-F, and RSV-N) and LAMP1 revealed that LAMP1 colocalized with RSV-F protein, but not with RSV-G and RSV-N proteins (Fig. 1a−f). To confirm these results, we further used the lysosomal extraction kit to extract lysosomes from RSV-infected cells (HEp-2 and HBECs) and detected the level of RSV-F protein in lysosomes. As shown in Fig. 1g, western blotting analysis confirmed the enrichment of RSV-F protein in lysosomes. Next, intracellular cholesterol levels were detected using the Amplex™ Red Cholesterol Assay Kit. As expected, RSV infection of HEp-2 cells significantly increased cholesterol content in a time-dependent manner compared to that in mock-infected cells (Fig. 1h). Thus, these results demonstrate that RSV infection of cells reprograms lysosomal cholesterol metabolism to promote RSV-F accumulation.

### RSV infection blocks cholesterol egress from lysosomes by reducing LAL activity

Cholesterol egress from lysosomes consists of the following two processes: (a) cholesterol transportation from the lysosomal lumen to the limiting membrane and (b) cholesterol egress from the limiting membrane to other cellular compartments[10]. Since LAL controls cholesterol transportation from the lysosomal lumen to the limiting membrane[12–14], we initially investigated whether RSV infection blocks cholesterol egress from lysosomes by regulating LAL activity. As shown in Fig. 2a, b, when orlistat, a lipase inhibitor[44], was applied as a positive control, LAL activity was significantly reduced. As expected, RSV infection, similar to orlistat treatment, inhibited LAL activity. Moreover, RSV infection slightly increased *LAL* gene transcription and protein expression in infected cells (Fig. 2c−e). Second, NPC1 and NPC2 also serve as key factors in regulating cholesterol egress from lysosomes[15–18], and their protein levels were examined using western blotting analysis. The results showed that the protein levels of NPC1 and NPC2 in RSV-infected cells significantly increased in a time-dependent manner compared with those in the mock-infected cells (Supplementary Fig. 2). The dynamic changes in NPC1 and NPC2 were consistent with the observed trend in lysosomal cholesterol content. The increase in lysosomal cholesterol content may be the main reason for the increase in NPC1 and NPC2 levels. Together, these results suggest that RSV infection blocks cholesterol egress from lysosomes by reducing LAL activity.

### RSV infection promotes LDL uptake by activating the SREBP2−LDLR axis

SREBP2 is a member of the SREBP family and mainly regulates cholesterol metabolism in cells[45]. The SREBP2 protein is normally localized in the ER in the form of an inactive precursor (pre-SREBP2). Under low-cholesterol conditions in the ER, SREBP2 is activated and cleaved by the S1P and S2P enzymes to generate an active fragment (nSREBP2)[22,23]. nSREBP2 translocates to the nucleus to recognize and bind to the SRE sequence[24], thereby initiating the transcription of LDLR and LDL uptake[25,26]. Since RSV infection blocks cholesterol transport from lysosomes to the ER, we examined whether RSV activates the SREBP2−LDLR axis. As shown in Fig. 3a−c, RSV induced the conversion of the inactive precursor (pre-SREBP2) to the active fragment (nSREBP2) in a time-dependent manner. An optimized dual-luciferase reporter assay further confirmed the effect of RSV infection on the transcriptional activity of SREBP2. The structure of the luciferase reporter plasmid constructed with a promoter containing the SRE sequence as the template is shown in Fig. 3d. RSV-infected cells

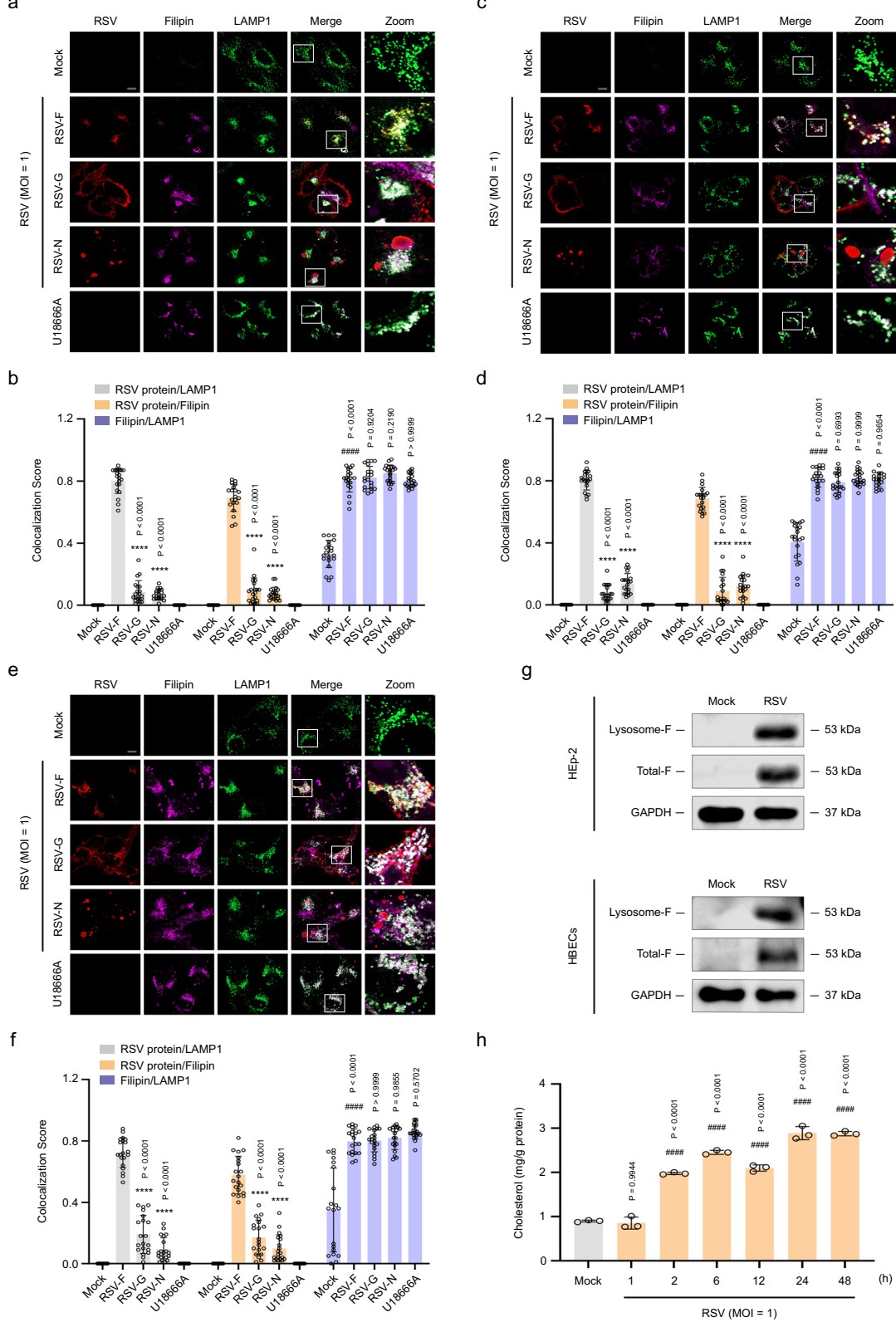

**Fig. 1 | RSV-F protein accumulates in cholesterol-rich lysosomes in infected cells.** HEp-2, 16HBE, or HBECs cells were either mock-infected or infected with RSV (MOI = 1) in the presence or absence of U18666A (10 μM) for the indicated durations. **a**–**f** Immunocolocalization of RSV proteins (RSV-F, RSV-G, and RSV-N), cholesterol (filipin III), and LAMP-1 in mock-infected, RSV-infected (24 h post-infection), and U18666A-treated cells (HEp-2 (**a**, **b**); 16HBE (**c**, **d**); HBECs (**e**, **f**). Scale bar: 10 μm. Data (n = 20 micrographs) are one representative of three independent experiments. **g** Western blotting analysis of RSV-F protein in lysosomes. **h** The cholesterol content in HEp-2 cells 0 h, 1 h, 2 h, 6 h, 12 h, 24 h, and 48 h after RSV infection was determined using an Amplex™ Red Cholesterol Assay Kit (n = 3 independent experiments). Image parameters: Scaling-per Pixel (**a**, **c**, **e**: 0.032 × 0.032 μm²); Image size-pixels (**a**, **c**, **e**: 2432 × 2432); image size-scaled (**a**, **c**, **e**: 78.01 × 78.01 μm²); objective (**a**, **c**, **e**: plan-apochromat 63×/1.40 oil DIC M27); scan zoom (**a**, **c**, **e**: 1.3). Data are shown as the mean ± SD, statistical analysis using one-way ANOVA (####P < 0.0001 compared to the blank control group; ****P < 0.0001 compared to the viral control group).

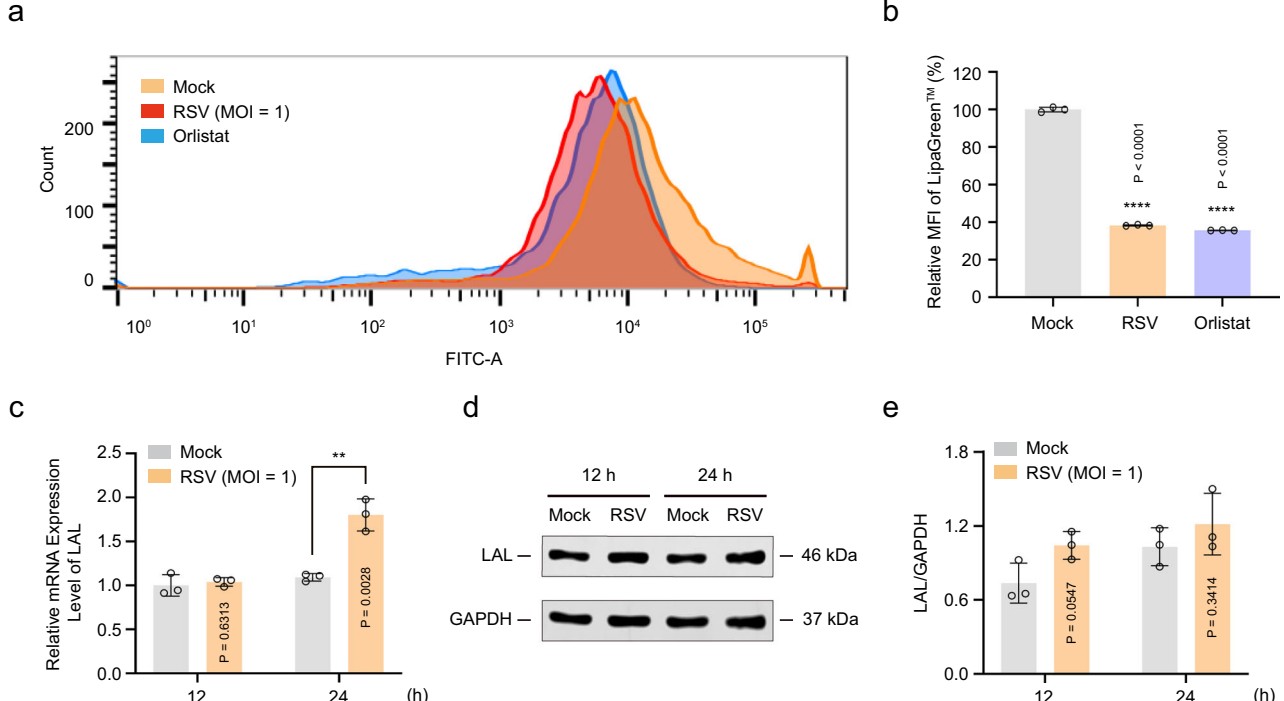

**Fig. 2 | RSV infection blocks cholesterol egress from lysosomes by reducing LAL activity.** HEp-2 cells were either mock-infected or infected with RSV (MOI = 1) in the presence or absence of orlistat (10 µM) for the indicated durations. **a, b** The activity of LAL in HEp-2 cells 24 h after RSV infection was determined using flow cytometry (*n* = 3 independent experiments). **c** The mRNA level of *LAL* gene in HEp-2 cells 12 h or 24 h after RSV infection was determined using RT-PCR (*n* = 3 independent experiments). **d, e** Western blotting analysis of LAL in HEp-2 cells 12 h or 24 h after RSV infection (*n* = 3 independent experiments). Data are shown as the mean ± SD, statistical analysis using two-sided Student's *t*-test (**c**, **e**) or one-way ANOVA (**b**) (**P < 0.01 and ****P < 0.0001 compared to the blank control group).

showed significantly higher firefly luciferase activity than mock-infected cells in a time-dependent manner. Correspondingly, RSV infection of HEp-2 cells significantly increased *LDLR* gene transcription and protein expression (Fig. 3e–g).

Cholesterol is a fat-soluble component that is insoluble in blood, body fluids, and culture media. In organisms or in vitro cell cultures, exogenous cholesterol needs to adhere to LDL and enter cells via endocytosis mediated by LDLR[11]. To further confirm that RSV infection promotes the uptake of exogenous cholesterol by host cells, DiI-labeled LDL (DiI-LDL) was added to the medium to simulate cellular uptake of exogenous cholesterol, and the fluorescence intensity of DiI-LDL was detected using flow cytometry. As shown in Fig. 3h, i, there was a 6-fold increase of DiI-LDL fluorescence intensity in the U18666A group compared with that in the mock-infected group, with U18666A acting as a positive control for DiI-LDL uptake. Consistent with the effects of U18666A, RSV infection also promoted cellular uptake of DiI-LDL in a time-dependent manner. Next, we used the Abberior STEDYCON to confirm the relationship between exogenous cholesterol and lysosomes following RSV infection. STED images revealed that DiI-LDL was completely wrapped in the lysosomal lumen in three cell lines (Fig. 3j–l), which is consistent with the above results (Fig. 1a–f and Fig. 2). Thus, these results demonstrate that RSV infection promotes LDL uptake by activating the SREBP2–LDLR axis.

## RSV-induced cholesterol accumulation in lysosomes inhibits autophagy flux in infected cells

Since lysosomal cholesterol metabolism is involved in the regulation of autophagy in host cells[27], the LC3 and p62/SQSTM1 levels were initially tested using western blotting analysis to investigate the effect of RSV on autophagy flux. As shown in Supplementary Fig. 3, consistent with the results using U18666A and chloroquine (CQ, an autophagy inhibitor)[46], RSV infection of HEp-2, 16HBE, or HBECs cells induced the conversion of LC3-I to LC3-II and increased p62 levels in a time-

dependent manner, indicating that RSV infection blocked autophagy flux.

## RSV-induced cholesterol accumulation in lysosomes controls minus-end transport of autophagosomes and autolysosome formation by altering ORP1L function

As a cholesterol sensor, ORP1L regulates the formation of ER–autophagosome/lysosome contact sites by interacting with VAP-A, which is essential for autophagosome localization and autolysosome formation[28]. The interaction of ORP1L with VAP-A is also cholesterol-sensitive. Under high-cholesterol conditions, the affinity of ORP1L for VAP-A decreases, which promotes minus-end transport of autophagosomes and autolysosome formation by inducing the formation of RILP–dynein, PLEKHM1–HOPS VPS39, and RILP–HOPS VPS41 complexes[28,47–51]. Since RSV induces lysosomal cholesterol accumulation and high cholesterol levels in infected cells, we speculated that RSV infection controls autophagosome localization and autolysosome formation by regulating the function of ORP1L. To verify this hypothesis, we initially constructed mCherry-tagged wild-type ORP1L and variants that alter its cholesterol-sensing properties. Of the latter, ORP1LΔORD lacks the cholesterol-interacting ORD domain and cannot be inactivated by cholesterol, thereby mimicking the low-cholesterol condition[52]. Two inactivating point mutations (Y477A and D478A) in the FFAT motif were introduced in ORP1LΔORD (ORP1LΔORD YDAA); the modification eliminates the VAP-A-binding site FFAT and restores the function of ORP1L in regulating the autophagosome even under conditions of low cholesterol[28] (Fig. 4a). Confocal microscopy images showed obvious colocalization of mCherry-tagged ORP1L and EGFP-tagged VAP-A in the mock, mock (Chol−), and ORP1LΔORD groups. mCherry-tagged ORP1L failed to interact with EGFP-tagged VAP-A following RSV infection. Treatment with U18666A and ORP1LΔORD YDAA as positive controls was consistent with the effects in RSV-infected cells. In contrast, depletion of cholesterol using MβCD effectively

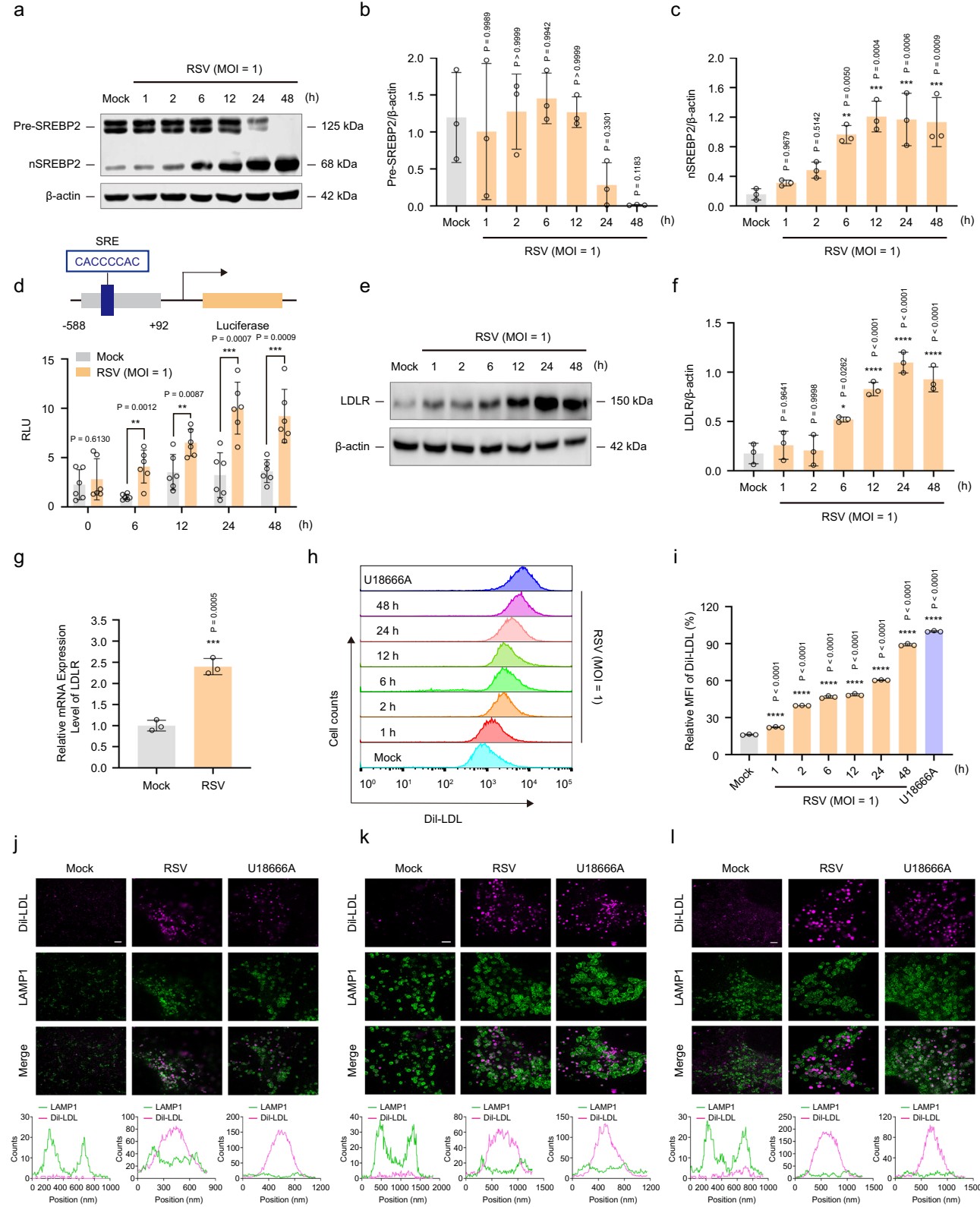

restored the colocalization of ORP1L and VAP-A in infected cells (Fig. 4b, c). Due to the ability of ORP1L to control the RILP–dynein complex[52], we investigated whether RSV infection triggers dynein–dynactin recruitment by RILP. Unlike the mock, mock (Chol−), and ORP1LΔORD groups, RSV infection significantly induced recruitment of the dynein–dynactin subunit p150^Glued to RILP. Consistently, U18666A and ORP1LΔORD YDAA also induced colocalization of EGFP-

tagged p150^Glued and BFP-tagged RILP, while a RILP mutant (RILPΔN)[53,54] that fails to recruit the dynein–dynactin motor released p150^Glued from the ORP1L–RILP complex (Fig. 4d–f). Corresponding to the above results, overexpression of ORP1L, ORP1LΔORD YDAA, or RILP in HEK293T cells induced marked accumulation of LC3B at the perinuclear region, while LC3B was dispersed throughout the cytoplasm in normal, ORP1LΔORD-transfected, and RILPΔN-transfected

**Fig. 3 | RSV-induced cholesterol accumulation in lysosomes promotes LDL uptake by activating the SREBP2–LDLR axis.** HEK293T cells transiently expressing pGL3-SRE-LUC and pRL-TK, HEp-2, 16HBE, or HBECs cells were either mock-infected or infected with RSV (MOI = 1) in the presence or absence of U18666A (10 μM) for the indicated durations. **a–c** Western blotting analysis of SREBP2 in HEp-2 cells 0 h, 1 h, 2 h, 6 h, 12 h, 24 h, and 48 h after RSV infection (n = 3 independent experiments). **d** The transcriptional activity of SREBP2 in HEK293T cells 0 h, 6 h, 12 h, 24 h, and 48 h after RSV infection was measured via the dual-luciferase reporter assay system (n = 6 independent experiments). **e, f** Western blotting analysis of LDLR in HEp-2 cells 0 h, 1 h, 2 h, 6 h, 12 h, 24 h, and 48 h after RSV infection (n = 3 independent experiments). **g** The mRNA level of the *LDLR* gene in HEp-2 cells 24 h after RSV infection was measured using RT-PCR (n = 3 independent

experiments). **h, i** Exogenous cholesterol (DiI-LDL, 30 μg/mL) uptake in HEp-2 cells 0 h, 1 h, 2 h, 6 h, 12 h, 24 h, and 48 h after RSV infection was determined using flow cytometry (n = 3 independent experiments). **j–l** Immunocolocalization of DiI-LDL and LAMP-1 in mock-infected, RSV-infected (24 h post-infection), and U18666A-treated cells (HEp-2 (**j**); 16HBE (**k**); HBECs (**l**)). Scale bar: 2 μm. Data are one representative of three independent experiments. Image parameters: pixel size (**j, l**: 20 nm; **k**: 25 nm); image size (**j, l**: 20 × 20 μm², 1000 × 1000 px²; **k**: 20 × 20 μm², 800 × 800 px²); Objective (**j, k, l**: 100×, 1.45). Data are shown as the mean ± SD, statistical analysis using two-sided Student's *t*-test (**d, g**) or one-way ANOVA (**b, c, f, i**) (*P < 0.05, **P < 0.01, ***P < 0.001, and ****P < 0.0001 compared to the blank control group).

HEK293T cells (Fig. 4g). Unexpectedly, knockdown of ORP1L with si-ORP1L or MβCD treatment had no effect on RSV-induced autophagosome migration in infected cells (Fig. 4h, i). Taken together, these results suggest that RSV-induced cholesterol accumulation in lysosomes contributes to the formation of the RILP–dynein complex by regulating ORP1L, thereby promoting minus-end transport of autophagosomes.

To further investigate whether RSV infection promotes autolysosome formation by regulating ORP1L, PLEKHM1 was co-expressed with ORP1L, Rab7, RILP, or HOPS VPS39 in HEK293T cells, and their interactions were detected using immunofluorescence assays. As shown in Fig. 5a–c, BFP-tagged PLEKHM1 failed to interact with EGFP-tagged Rab7 in the mock, mock (Chol−), ORP1LΔORD, and Rab7 T22N (dominant-negative Rab7)[53] groups. In contrast, there was obvious colocalization between EGFP-tagged Rab7 and BFP-tagged PLEKHM1 following RSV infection. Application of U18666A, ORP1LΔORD YDAA, and Rab7 Q67L (constitutively active Rab7)[53] as positive controls was consistent with the results in RSV-infected cells. Depletion of cholesterol using MβCD effectively blocked colocalization between Rab7 and PLEKHM1 in infected cells. Second, the puncta showing that PLEKHM1 colocalized with HOPS VPS39 also markedly increased after RSV infection, which was abolished by MβCD (Fig. 5d, e). Third, RSV-induced cholesterol accumulation in lysosomes did not affect the interaction between RILP and PLEKHM1/HOPS VPS41 (Supplementary Fig. 4). Corresponding to the above results, overexpression of ORP1L, ORP1LΔORD YDAA, RILP, Rab7, or Rab7 Q67L in HEK293T cells resulted in colocalization between p62 and LAMP1 compared with the control, ORP1LΔORD, RILPΔN, and Rab7 T22N groups (Fig. 5f–i). Lastly, the results of colocalization of p62 and LAMP1 in HEp-2 or 16HBE cells showed that p62-positive granules were uniformly distributed in the cytoplasm in the mock and mock (Chol−) groups, and most of the p62 accumulated and colocalized with the LAMP1 puncta after exposure to RSV or U18666A. Unexpectedly, colocalization of p62 and LAMP1 induced by RSV was not abolished in si-ORP1L-transfected and MβCD-treated cells (Fig. 5j–o). Taken together, these results suggest that RSV-induced cholesterol accumulation in lysosomes contributes to the formation of the Rab7–PLEKHM1–HOPS VPS39 complex by regulating ORP1L, thereby promoting autolysosome formation.

### RSV-induced cholesterol accumulation in lysosomes inhibits autolysosome degradation in infected cells

Since cholesterol accumulation is closely associated with lysosomal dysfunction[29], we speculated that RSV-induced cholesterol accumulation in lysosomes inhibits autolysosome degradation in infected cells. To verify this hypothesis, we initially exploited a mRFP-GFP-LC3B tandem sensor to ascertain lysosomal acidification. This sensor contains an acid-sensitive GFP mutant (fluorescence quenching at acidic pH) and an acid-insensitive RFP (fluorescing at both acidic and neutral pH)[55]. Its use allows monitoring of the progression from autophagosome to autolysosome or lysosome pH based on the intensity of the GFP signal. As shown in Fig. 6a, rapamycin (RAPA), an autophagy activator[56], dramatically decreased the GFP signal and increased the

RFP signal because it promoted autophagy flux without affecting the pH of lysosomes. Consistent with the effects of CQ (inhibitor of lysosome acidification)[46] and U18666A, both GFP and RFP fluorescence signals were greater in RSV-infected cells than in mock-infected cells, indicating accumulation of autophagosomes and inhibition of lysosomal acidity during RSV infection. MβCD treatment abolished the RSV-induced increases in the GFP signal. A LysoSensor staining assay further confirmed the effect of RSV-induced cholesterol retention in lysosomes on acidification. In this assay, the LysoTracker probe specifically labels the lysosome. The LysoSensor probe binds to acidic organelles through protonation, and its signal is enhanced with the acidification of organelles, which can be used to indicate lysosomal pH[57]. The LysoSensor signal was significantly attenuated following RSV infection or CQ treatment. Lysosomal acidity in infected cells (HEp-2, 16HBE, and HBECs) was rescued only by the depletion of cholesterol (Fig. 6b–g). Since lysosomal acidification is the primary condition that ensures the function of its internal hydrolase, we utilized flow cytometry to measure lysosomal activity during RSV infection. The results showed a significant reduction in lysosomal activity in the RSV-infected, U18666A-treated, and CQ-treated groups compared with that in the mock-infected group. Treatment with bafilomycin A1 (BafA1, a cell-permeable inhibitor of endocytosis)[58] as a positive control was consistent with the results in RSV-infected cells. Cholesterol depletion largely reversed the effect of RSV on lysosomal activity (Fig. 6h, i). These data suggest that RSV-induced cholesterol retention in lysosomes impairs acidification and activity. Lastly, we utilized transmission electron microscopy (TEM) to detect the number of autolysosomes in infected cells (HEp-2, 16HBE, and HBECs) and discovered that RSV-infected cells accumulated more autolysosomes than mock-infected cells (Fig. 6j–o). Collectively, these data demonstrate that RSV-induced cholesterol accumulation in lysosomes inhibits autolysosome degradation, thereby blocking autophagy flux.

### Depletion of LDLR inhibits RSV infection in vitro and in vivo

To confirm the role of lysosomal cholesterol metabolism in RSV infection, we used small interfering RNA (siRNA) to knockdown the core factor, LDLR, and performed a series of virological experiments to determine whether LDLR controls RSV infection. As shown in Fig. 7a–f, the LDLR-specific siRNA significantly inhibited LDLR expression in three cell lines, as detected using western blotting, and the yield of progeny virions in HEp-2, 16HBE, or HBECs cells was only 24.41%, 9.87%, or 7.27% of that of the control, respectively. Nearly 100% of the cells remained viable at si-LDLR concentrations up to 50 nM, demonstrating an appreciable Therapeutic Index (TI) and suggesting that the anti-RSV activity is not due to cytotoxicity. Therefore, these results suggest that LDLR plays a critical role in RSV infection of host cells.

To determine whether LDLR affects RSV infection by regulating RSV-F protein, we examined the mRNA and protein expression of RSV *F* gene 6 h, 12 h, or 24 h after RSV infection using RT-PCR and western blotting analysis, respectively. As shown in Fig. 7g, there was no significant difference in the mRNA level of RSV *F* between the si-LDLR and RSV groups 6 h, 12 h, and 24 h after RSV infection, suggesting that

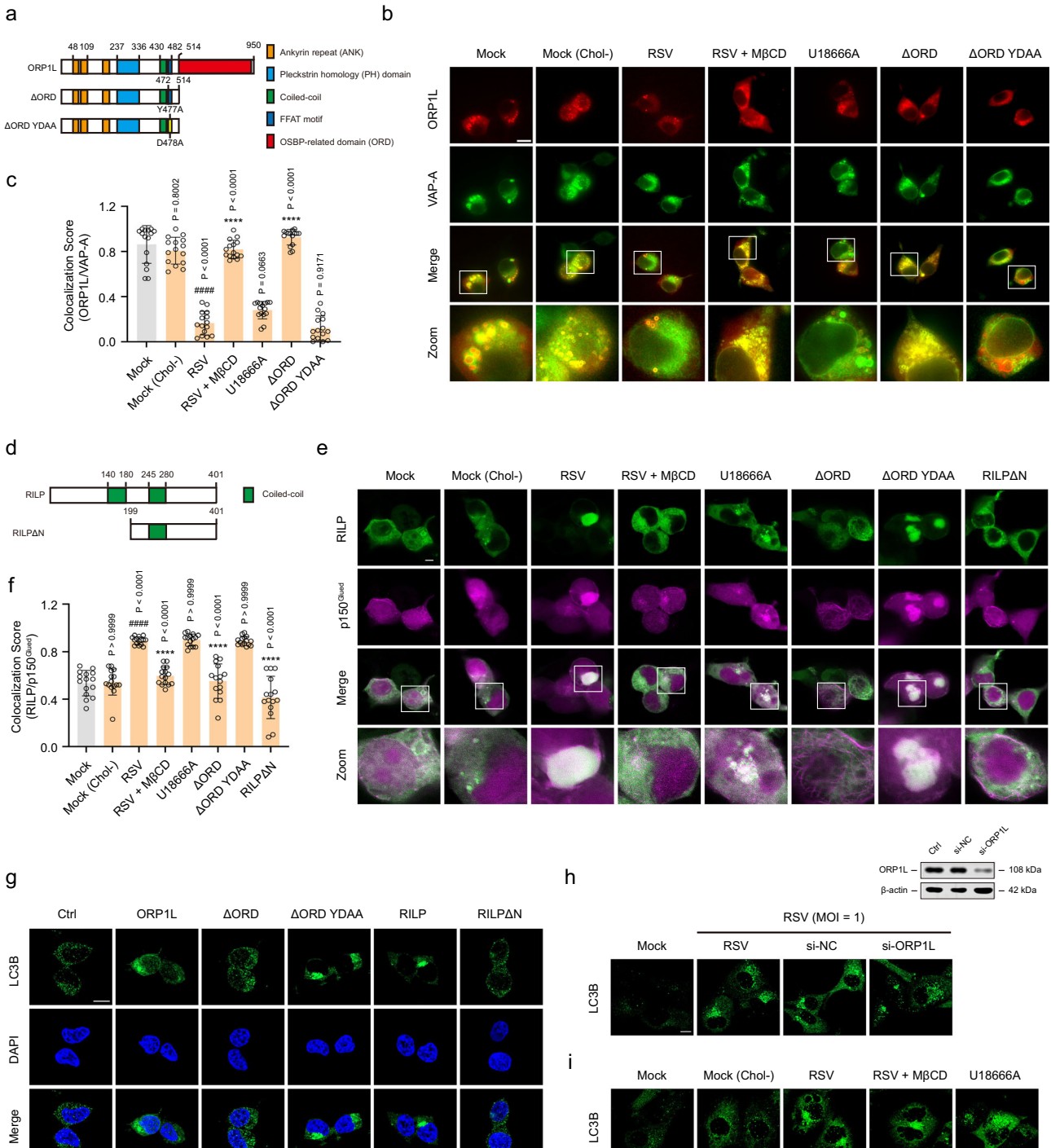

**Fig. 4 | RSV-induced cholesterol accumulation in lysosomes promotes minus-end transport of autophagosomes by regulating ORP1L.** HEK293T cells transiently expressing the indicated plasmids or HEp-2 cells were either mock-infected or infected with RSV (MOI = 1) in the presence or absence of si-NC (negative control nucleotide, 50 nM), si-ORP1L (50 nM), U18666A (10 μM), or MβCD (100 μM) for 24 h. **a**–**c** Immunocolocalization of ORP1L and VAP-A in infected HEK293T cells (*n* = 15 micrographs). Scale bar: 10 μm. **d**–**f** Immunocolocalization of RILP and p150^Glued in infected HEK293T cells (*n* = 15 micrographs). Scale bar: 10 μm. **g** The effect of ORP1L or RILP on LC3B localization in HEK293T cells was detected using an immunofluorescence assay. Scale bar: 10 μm. **h** The effect of si-ORP1L or si-NC on LC3B localization in infected HEp-2 cells was detected using an

immunofluorescence assay. Scale bar: 10 μm. **i** Immunofluorescence analysis of LC3B localization in infected HEp-2 cells. Scale bar: 10 μm. Image parameters: scaling-per pixel (**b**: 0.031 × 0.031 μm²; **e**: 0.025 × 0.025 μm²; **h**, **i**: 0.032 × 0.032 μm²); image size-pixels (**b**: 1024 × 1024; **e**: 2048 × 2048; **h**, **i**: 2432 × 2432); image size-scaled (**b**: 64.00 × 64.00 μm²; **e**: 50.71 × 50.71 μm²; **h**, **i**: 78.01 × 78.01 μm²); objective (**b**: 63 × 1.40NA oil objective; **e**, **h**, **i**: plan-apochromat 63×/1.40 oil DIC M27); scan zoom (**b**: 1.6; **e**: 2.0; **h**, **i**: 1.3). Data are one representative of three independent experiments and are shown as the mean ± SD, statistical analysis using one-way ANOVA (^####*P* < 0.0001 compared to the blank control group; ****P* < 0.0001 compared to the viral control group).

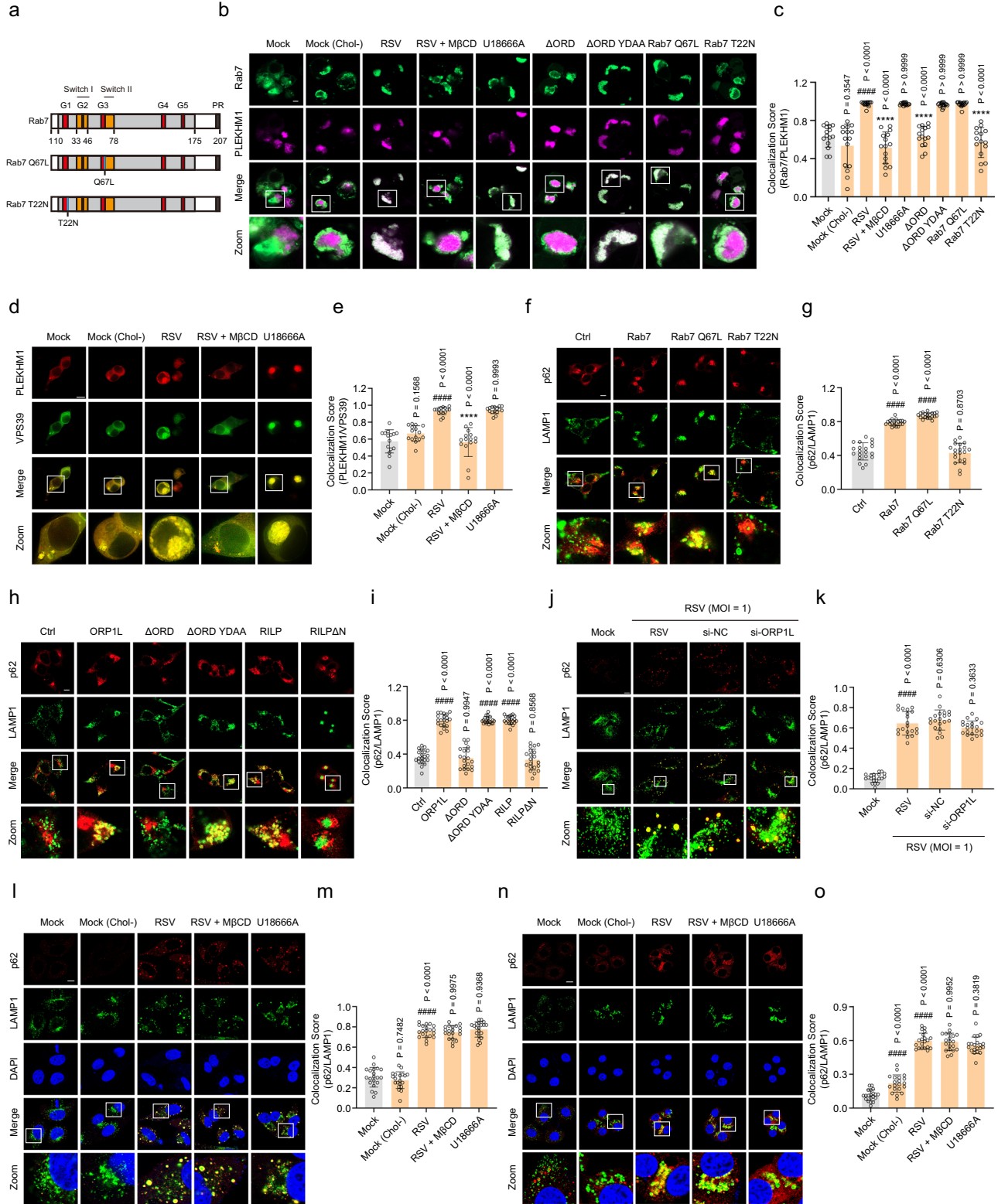

LDLR does not affect RSV *F* gene transcription. In contrast, si-LDLR significantly reduced RSV-F protein content, and the inhibition rates in HEp-2, 16HBE, and HBECs cells were 68.88%, 84.19%, and 87.86%, respectively. The increase in cholesterol content caused by RSV was also reversed in LDLR-knockdown cells. Interestingly, supplementation of exogenous cholesterol restored RSV-F protein content in LDLR-knockdown cells, but did not appreciably affect infection of control cells (Fig. 7h–l). To further determine whether the hiding and accumulation of RSV-F protein is regulated by LDLR-mediated exogenous

cholesterol uptake rather than endogenous cholesterol synthesis, the following methods were used: (a) blockade of exogenous cholesterol uptake with lipid-free medium; and (b) blockade of cellular endogenous cholesterol synthesis with lovastatin, a specific inhibitor of 3-hydroxy-3-methylglutaryl coenzyme A (HMG-CoA) reductase (HMGCR, a rate-limiting enzyme in the synthesis of endogenous cholesterol)[59]. As shown in Fig. 7m–o, the blockade of exogenous cholesterol uptake significantly inhibited RSV-F protein accumulation and the up-regulation of cholesterol content in RSV-infected cells.

**Fig. 5 | RSV-induced cholesterol accumulation in lysosomes promotes auto-lysosome formation by regulating ORP1L.** HEK293T cells transiently expressing the indicated plasmids, HEp-2, or 16HBE cells were either mock-infected or infected with RSV (MOI = 1) in the presence or absence of si-NC (50 nM), si-ORP1L (50 nM), U18666A (10 μM), or MβCD (100 μM) for 24 h. **a–c** Immunocolocalization of Rab7 and PLEKHM1 in infected HEK293T cells (n = 15 micrographs). Scale bar: 10 μm. **d**, **e** Immunocolocalization of PLEKHM1 and VPS39 in infected HEK293T cells (n = 15 micrographs). Scale bar: 10 μm. **f–i** The effect of Rab7, ORP1L, or RILP on the colocalization of p62 and LAMP1 in HEK293T cells was detected using an immunofluorescence assay (n = 20 micrographs). Scale bar: 10 μm. **j**, **k** The effect of si-ORP1L or si-NC on the colocalization of p62 and LAMP1 in infected HEp-2 cells was detected using an immunofluorescence assay (n = 20 micrographs). Scale bar:

10 μm. **l–o** Immunocolocalization of p62 and LAMP1 in infected cells (HEp-2: **l**, **m**; 16HBE: **n**, **o**) (n = 20 micrographs). Scale bar: 10 μm. Image parameters: scaling-per pixel (**b**: $0.025 \times 0.025$ μm²; **d**, **j**: $0.031 \times 0.031$ μm²; **f**, **h**: $0.028 \times 0.028$ μm²; **l**, **n**: $0.032 \times 0.032$ μm²); image size-pixels (**b**: $2048 \times 2048$; **d**: $1024 \times 1024$; **f**, **h**, **l**, **n**: $2432 \times 2432$; **j**: $1536 \times 1536$); image size-scaled (**b**: $50.71 \times 50.71$ μm²; **d**: $64.00 \times 64.00$ μm²; **f**, **h**: $67.61 \times 67.61$ μm²; **l**, **n**: $78.01 \times 78.01$ μm²; **j**: $96.00 \times 96.00$ μm²); objective (**b**, **f**, **h**, **n**: plan-apochromat 63×/1.40 oil DIC M27; **d**, **j**: 63 × 1.40 NA oil objective); scan zoom (**b**: 2.0; **d**, **j**: 1.6; **f**, **h**: 1.5; **l**, **n**: 1.3). Data are one representative of three independent experiments and are shown as the mean ± SD, statistical analysis using one-way ANOVA (####P < 0.0001 compared to the blank control group; ****P < 0.0001 compared to the viral control group).

However, lovastatin treatment of cells had little effect on these events. RSV also failed to alter HMGCR content in infected cells (Supplementary Fig. 5). Collectively, these data demonstrate that lysosomal cholesterol metabolism is required for RSV-F accumulation.

LDLR-knockout C57BL/6 mice (LDLR$^{-/-}$) were used to further verify the effect of LDLR on RSV infection in vivo. As shown in Fig. 8a, the viral load in the LDLR$^{-/-}$ group was log$_{10}$ 2.14 PFU g$^{-1}$ and was remarkably lower than that in the RSV group (log$_{10}$ 3.45 PFU g$^{-1}$). The results of the immunofluorescence staining and RT-PCR assay indicated that knockout of LDLR also reduced the total amount of RSV protein and RSV *NS1* and *NS2* genes (Fig. 8b–d). Moreover, mice challenged with RSV showed severe inflammatory responses, including (a) higher lung indices; (b) aberrant elevation of leukocytes (WBC), neutrophils (NE), lymphocytes (LY), and monocytes (MO) in the blood; and (c) virus-associated alveolar damage, pulmonary inflammatory infiltration, and bronchopneumonia. Knockout of LDLR significantly reversed these effects (Fig. 8e–j). Together, our results provide strong evidence that knockout of LDLR effectively protects mice from RSV challenges by blocking virion replication and virus-induced pneumonia.

## Discussion

Increasing evidence suggests that viruses have evolved multiple mechanisms to reprogram the cellular metabolism of the host for optimal replication[60–62]. However, various viruses participate in host metabolic regulation through different mechanisms, and not all infections result in abnormal lysosomal cholesterol metabolism. Our data provide new insights into the mechanism by which RSV infection of host cells triggers cholesterol reprogramming. We found that RSV infection blocks exogenous cholesterol egress from lysosomes by retaining it in the lysosomal lumen, which is achieved by impairing the function of LAL instead of NPC1 or NPC2. Since the transport of exogenous cholesterol is interrupted and cholesterol fails to reach the ER, the cholesterol content in ER is significantly reduced. The closed structure of Insig and SCAP proteins opens, and the SREBP2–SCAP complex is released from the ER and translocates to the Golgi apparatus[63]. SREBP2 is subsequently hydrolyzed to form nSREBP2. The latter enters the nucleus through nuclear pores, recognizes and binds to the SRE sequence to initiate the transcription of the cholesterol uptake receptor LDLR, and finally upregulates the uptake of LDL-containing exogenous cholesterol. This is the main reason for the activation of SREBP2 and up-regulation of LDLR after RSV infection of host cells. Correspondingly, exogenous cholesterol is continuously taken up by infected cells and accumulates in lysosomes, eventually resulting in abnormal increases in cholesterol levels in infected cells.

Cholesterol metabolism can regulate autophagy and lysosomes in host cells. Cholesterol storage leads to enlarged lysosomes that exhibit functional, trafficking, and morphological defects, which inhibits the degradation of autophagosomes[64,65]. On the other hand, cholesterol is also involved in the transport and maturation of autophagosomes and autolysosome formation[28]. In this study, we elucidated how RSV regulates autophagy by reprogramming lysosomal cholesterol

metabolism. First, RSV-induced cholesterol accumulation in lysosomes induces autophagosome transport and localization to the perinuclear region. The reason for this phenomenon is that the high-cholesterol state configuration of ORP1L loses its binding activity to VAP-A and promotes the recruitment of the dynein–dynactin subunit p150$^{Glued}$ to RILP, thereby facilitating minus-end transport of autophagosomes. RSV-induced lysosomal cholesterol reprogramming also contributes to autolysosome formation, which is achieved through ORP1L-mediated formation of the Rab7–PLEKHM1–HOPS VPS39 complex. Although the RILP–HOPS VPS41 complex is also an important regulator of autolysosome formation, our results confirmed that the regulation of autophagy by RSV is not dependent on the RILP–HOPS VPS41 complex. Surprisingly, although we confirmed the role of RSV-mediated lysosomal cholesterol reprogramming in autophagosome transport and autolysosome formation, the use of MβCD to eliminate cholesterol did not abolish these autophagic events during RSV infection. Combined with previous studies[66,67], the possible reason is the existence of other cellular factors or signaling pathways that regulate autophagy in addition to lysosomal cholesterol metabolism during RSV infection. However, this remains to be further explored. Second, as cholesterol retention in lysosomes results in acidification inhibition, the function of hydrolase in the lumen is lost, which is the main reason for the inactivation of LAL. Correspondingly, the hydrolytic function of lysosomes is impaired, and autophagosomes accumulate in lysosomes without being degraded. Cholesterol depletion or MβCD treatment mobilizes accumulated cholesterol in lysosomes of RSV-infected cells, restores lysosome function, and promotes autophagy flux. This was further validated by blocking autophagy flux with CQ (inhibitor of lysosomal acidification) and U18666A (inhibitor of lysosomal cholesterol efflux). In summary, the impairment of autophagy flux is associated with lysosomal dysfunction as a consequence of cholesterol accumulation in RSV-infected cells, and the regulation of autophagy by RSV infection manifests as an abnormal increase in autophagic activity and a secondary reduction in autophagy flux caused by the impairment of autolysosomes.

The F protein is a glycoprotein located on the surface of the virus and is essential for RSV replication. First, the F protein facilitates the fusion of the viral membrane with the host-cell membrane, allowing the viral nucleic acid into the cytoplasm[68]. Second, expression of RSV-F protein on the surface of infected cells can also trigger fusion with neighboring cells, which promotes the spread of progeny viruses between cells[69–71]. In this study, we found that RSV-F protein is stored in cholesterol-rich lysosomes, indicating that RSV-mediated cholesterol reprogramming causes lysosomal dysfunction, providing a site for RSV-F accumulation. Cholesterol depletion or knockdown of LDLR greatly reduces the level of RSV-F protein but does not affect RSV *F* gene transcription in infected cells. The lack of RSV-F protein means that the entry of progeny viruses into cells and their spread between cells will be hindered. Consequently, the knockdown of LDLR effectively inhibits RSV infection in vitro. Interestingly, supplementation of exogenous cholesterol restores RSV-F protein content in LDLR-knockdown cells. In contrast, lovastatin treatment of cells has little

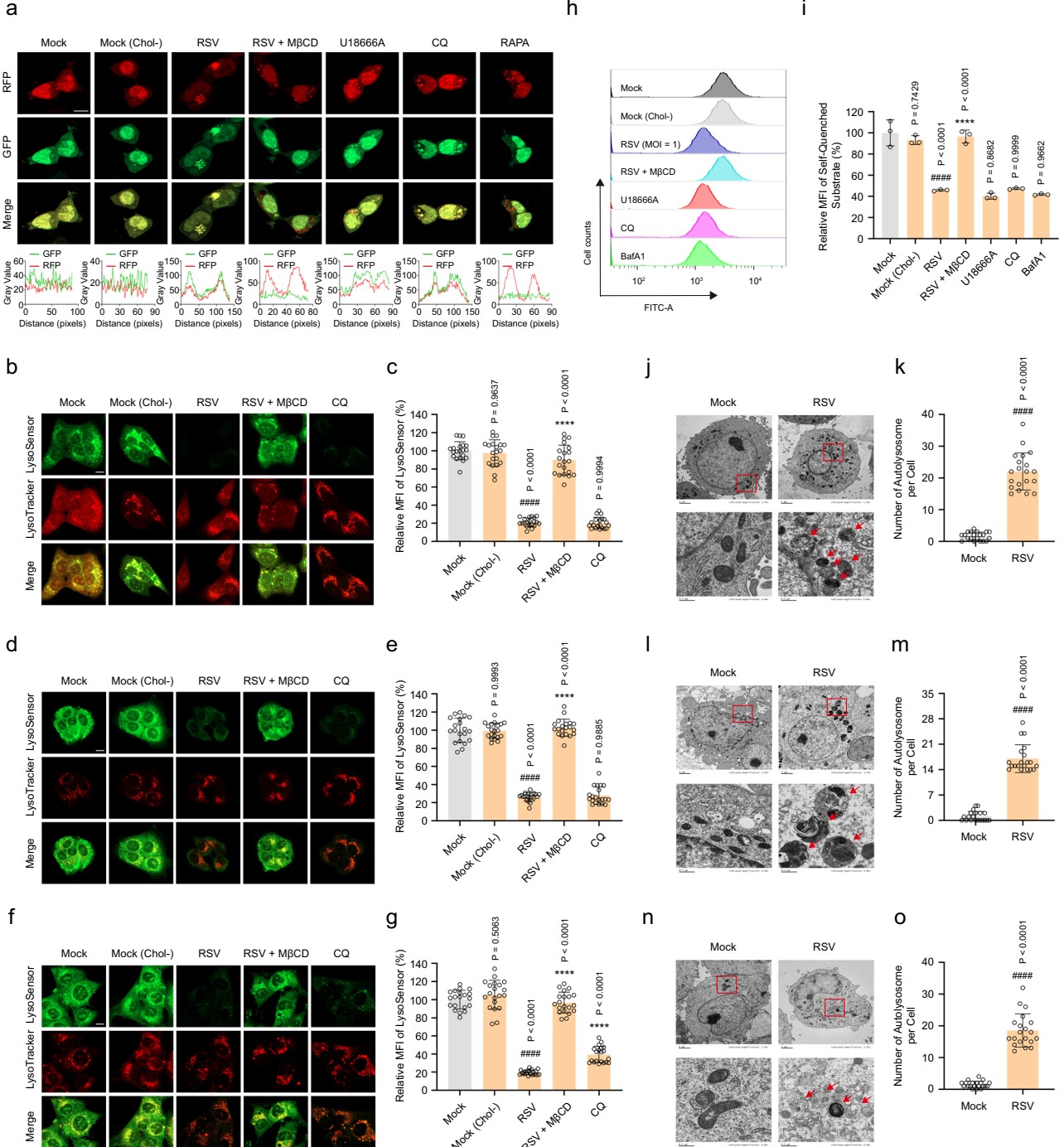

**Fig. 6 | RSV-induced cholesterol accumulation in lysosomes blocks autophagy flux by inhibiting autolysosome degradation.** HEK293T cells transiently expressing mRFP-GFP-LC3B, HEp-2, 16HBE, or HBECs cells were either mock-infected or infected with RSV (MOI = 1) in the presence or absence of U18666A (10 μM), CQ (5 μM), RAPA (200 nM), MβCD (100 μM), or BafA1 (200 nM) for 24 h. **a** The fluorescence intensity of mRFP-GFP-LC3B in infected HEK293T cells was measured using a confocal microscope with Airyscan. Scale bar: 10 μm. Data are one representative of three independent experiments. **b**–**g** The fluorescence intensity of LysoSensor in infected cells was measured using a confocal microscope with Airyscan (HEp-2 (**b**, **c**); 16HBE (**d**, **e**); HBECs (**f**, **g**)). Scale bar: 10 μm. Data (n = 20 micrographs) are one representative of three independent experiments. **h**, **i** The effect of RSV on lysosomal activity in infected HEp-2 cells was detected

using flow cytometry (n = 3 independent experiments). **j**–**o** The number of auto-lysosomes in infected cells was examined using TEM (HEp-2 (**j**, **k**); 16HBE (**l**, **m**); HBECs (**n**, **o**)). The red arrow indicates the autolysosome. Scale bar: 2 μm or 500 nm. Data (n = 20 cells) are one representative of three independent experiments. Image parameters: scaling-per pixel (**a**: 0.025 × 0.025 μm²; **b**, **d**, **f**: 0.038 × 0.038 μm²); image size-pixels (**a**, **b**, **d**, **f**: 2048 × 2048); image size-scaled (**a**: 50.71 × 50.71 μm²; **b**, **d**, **f**: 78.01 × 78.01 μm²); objective (**a**, **b**, **d**, **f**: plan-apochromat 63×/1.40 oil DIC M27); scan zoom (**a**: 2.0; **b**, **d**, **f**: 1.3). Data are shown as the mean ± SD, statistical analysis using two-sided Student's t-test (**k**, **m**, **o**) or one-way ANOVA (**c**, **e**, **g**, **i**) (####P < 0.0001 compared to the blank control group; ****P < 0.0001 compared to the viral control group).

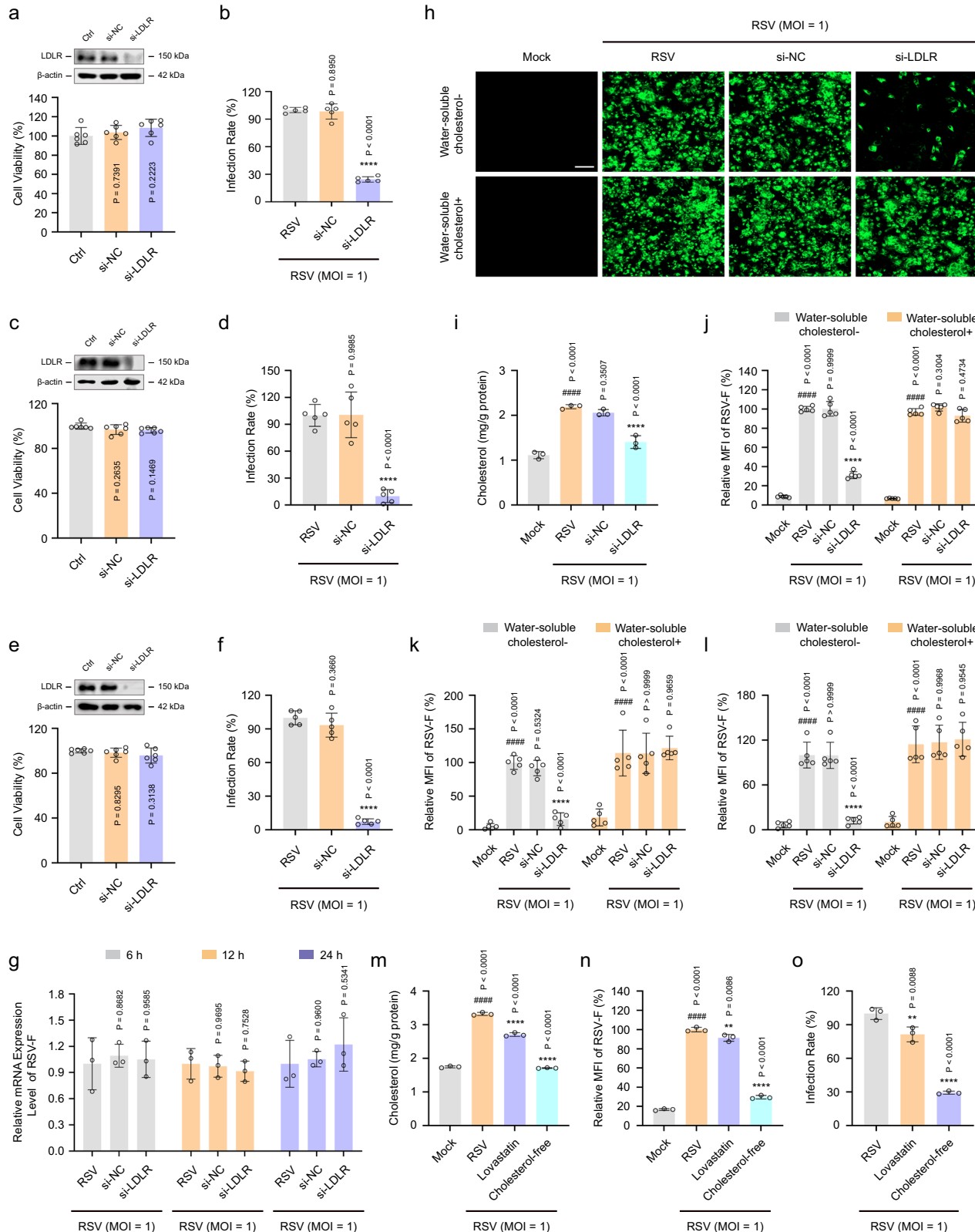

effect on RSV-F protein content, indicating that HMGCR-regulated synthesis of endogenous cholesterol is not essential for RSV replication. Furthermore, we established an RSV-infected mouse model via intranasal infection of C57BL/6 mice to verify the effect of LDLR on RSV infection in vivo. The results showed a large number of viral particles and severe inflammatory reactions in the lung tissues of the RSV-infected mice and abnormal increases in the number of inflammatory cells in the blood, indicating that RSV-induced pneumonia was successfully established. Most importantly, in vitro modulation of LDLR during RSV infection was mirrored in the RSV-infected mouse model. Knockout of LDLR significantly inhibits RSV multiplication in the lung tissues. The lung index, histopathological analysis, and analyses of five blood parameters indicated that knockout of LDLR also attenuates pneumonia symptoms and inflammation in RSV-infected mice.

**Fig. 7 | Knockdown of LDLR reduces RSV-F protein content by regulating exogenous cholesterol uptake in infected cells.** HEp-2, 16HBE or HBECs cells were mock-infected or infected with RSV (MOI = 1) and then incubated in the normal or cholesterol-free medium in the presence or absence of si-LDLR (50 nM), si-NC (50 nM), water-soluble cholesterol (40 μM), or lovastatin (25 μM) for the indicated durations. **a, c, e** Cytotoxicity of si-LDLR against cells was detected using a CCK-8 assay (HEp-2 (**a**); 16HBE (**c**); HBECs (**e**)) (*n* = 6 independent experiments). **b, d, f** The effect of si-LDLR on viral titers in RSV-infected cells was detected using an immunofluorescence assay (HEp-2 (**b**); 16HBE (**d**); HBECs (**f**)) (*n* = 5 independent experiments). **g** The effect of si-LDLR on the mRNA level of the RSV *F* gene in infected HEp-2 cells 6 h, 12 h, and 24 h after RSV infection was measured using RT-

PCR (*n* = 3 independent experiments). **h–l** The effect of si-LDLR on cholesterol content or protein level of RSV-F 24 h after RSV infection was determined using an Amplex™ Red Cholesterol Assay Kit or immunofluorescence assay, respectively (HEp-2 (**h–j**); 16HBE (**k**); HBECs (**l**)) (**h, j, k, l**: *n* = 5 independent experiments; **i**: *n* = 3 independent experiments). Scale bar: 100 μm. **m–o** The effect of exogenous or endogenous cholesterol on cholesterol content (**m**), the protein level of RSV-F (**n**), and viral titers (**o**) in infected HEp-2 cells was determined using an Amplex™ Red Cholesterol Assay Kit and immunofluorescence assay, respectively (*n* = 3 independent experiments). Data are shown as the mean ± SD, statistical analysis using one-way ANOVA (####*P* < 0.0001 compared to the blank control group; **\*\*P* < 0.01; and \*\*\*\**P* < 0.0001 compared to the viral control group).

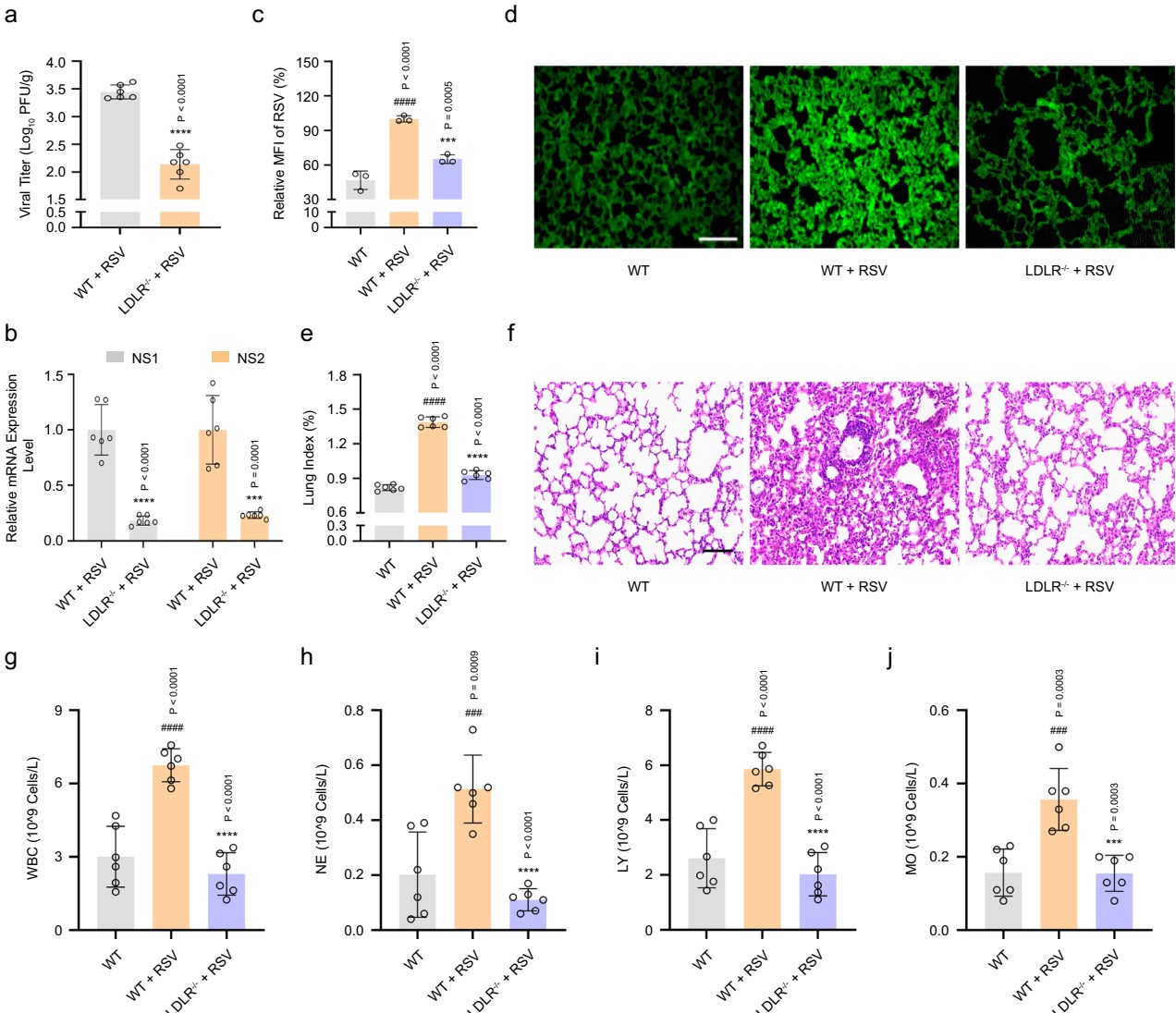

**Fig. 8 | Knockout of LDLR inhibits RSV infection in vivo.** WT or LDLR⁻/⁻ C57BL/6 mice were mock-infected or infected with RSV. On day 4 after RSV infection, the viral titers (**a**: *n* = 6 mice), RSV *NS1* and *NS2* mRNA levels (**b**: *n* = 6 mice), and RSV protein level (**c, d**: *n* = 3 mice) in the lung tissues of each group were measured using a plaque assay, RT-PCR, and immunofluorescence assay, respectively. Scale bar: 200 μm. **e** The lung indices represent the ratio of lung weight to body weight (*n* = 6 mice). **f** Histopathological analysis of the lung tissues in each group was carried out

using hematoxylin-eosin staining. Scale bar: 100 μm. **g–j** The number of WBC, NE, LY, and MO in the blood of each group was detected using a 5-part differential automated hematological analyzer (*n* = 6 mice). Data are shown as the mean ± SD, statistical analysis using two-sided Student's *t*-test (**a, b**) or one-way ANOVA (**c, e, g–j**) (###*P* < 0.001 and ####*P* < 0.0001 compared to the blank control group; \*\*\*P* < 0.001, and \*\*\*\**P* < 0.0001 compared to the viral control group).

Combined with the above results, targeting lysosomal cholesterol metabolism appears to be a promising approach to inhibit RSV infection.

   In summary, RSV-mediated cholesterol reprogramming is manifested by an increased uptake of exogenous cholesterol and retention

of cholesterol in lysosomes, which leads to lysosomal dysfunction and impaired autophagic flux. Cholesterol-rich lysosomes provide sites for RSV-F accumulation and facilitate RSV replication (Fig. 9). Importantly, LDLR, a crucial mediator of lysosomal cholesterol reprogramming, is required for RSV infection. Depletion of LDLR effectively inhibited RSV

infection both in vitro and in vivo. Therefore, this study presents new insights into RSV-mediated lysosomal cholesterol metabolism and suggests a target for the development of novel and highly efficient anti-RSV infection drugs.

# Methods

## Cells and virus

HEp-2 (ATCC CCL-23), 16HBE (ATCC CRL-9609), and HEK293T cells (ATCC CRL-3216) were purchased from the American Type Culture Collection (ATCC). HBEC cells (IMP-H041) were purchased from IMMOCELL (Xiamen, China). The RSV A2 (ATCC VR-1540) was obtained from the Medicinal Virology Institute of Wuhan University (Hubei, China). HEp-2, 16HBE, and HEK293T cells were cultured in Dulbecco's modified Eagle medium supplemented with 10% fetal bovine serum and 100 U/mL penicillin−streptomycin (PS). HBEC cells were cultured in a special culture medium for HBES (Cat# IMP-H041-1, IMMOCELL). RSV A2 was propagated in HEp-2 cells. The viral titer was determined using a plaque assay and viral stocks were stored at −80 °C.

## Compounds and antibodies

U18666A, orlistat, CQ, RAPA, BafA1, and lovastatin were obtained from MedChemExpress (Monmouth Junction, NJ, USA). MβCD was purchased from Macklin (Shanghai, China). Anti-RSV F glycoprotein (Cat# ab94968; 1:1000; LOT: GR3370956-1), anti-RSV G glycoprotein (Cat# ab94966; 1:1000; LOT: GR3247185-12), anti-RSV nucleoprotein (Cat# ab94806; 1:1000; LOT: 1029087-8), anti-calreticulin (Cat# ab92516; 1:500; LOT: GR3287998-11), anti-Niemann Pick C1 (Cat# ab134113; 1:2000; LOT: GR3230423-6), anti-Niemann Pick C2 (Cat# ab218192; 1:2000; LOT: GR3421834-3), anti-LDL receptor (Cat# ab52818; 1:1000; LOT: 1000292-1), anti-ORP1 (Cat# ab131165; 1:1000; LOT: GR3283699-8), anti-p62/SQSTM1 (Cat# ab109012; 1:400; LOT: 1000346-40), anti-HMGCR (Cat# ab242315; 1:1000; LOT: 1037157-2), anti-beta actin antibody (Cat# ab8226; 1:1000; LOT: GR3396181-2), goat anti-rabbit IgG H&L (Alexa Fluor® 488, Cat# ab150077, 1:1000, LOT: 1052444-22; Alexa Fluor® 594, Cat# ab150080, 1:1000, LOT: GR3373513-1; Alexa Fluor® 647, Cat# ab150079, 1:1000, LOT: 3444080-2), and goat anti-mouse IgG H&L (Alexa Fluor® 488, Cat# ab150113, 1:1000, LOT: GR3419505-1; Alexa Fluor® 594, Cat# ab150116, 1:1000, LOT: GR3438808-1) were purchased from Abcam (Cambridge, UK). Anti-LAMP1 (Cat# 9091S, 1:400, LOT: 6; Cat# 15665S, 1:100, LOT: 4), anti-LC3B (Cat# 83506S; WB, 1:1000; IF, 1:800; LOT: 2), anti-rabbit IgG, HRP-linked antibody (Cat# 7074; 1:2000; LOT: 32), and anti-mouse IgG, HRP-linked antibody (Cat# 7076; 1:2000; LOT: 38) were obtained from Cell Signaling Technology (Danvers, MA, USA). Anti-SREBP2 antibody (Cat# sc-13552; WB, 1:1000; IF, 1:50; LOT: GR3376305-1) was acquired from Santa Cruz Biotechnology (Dallas, TX, USA). Anti-LAL antibody (Cat# BF0079; 1:1000; LOT: 19c1337) was obtained from Affinity Biosciences (Changzhou, China). Anti-p62/SQSTM1 antibody (Cat# PM045; 1:400; LOT: 021) was purchased from MBL (Beijing, China). Anti-GAPDH antibody (Cat# A5028; 1:1000; LOT: GR3401390-1) was obtained from Bimake (Shanghai, China).

## Cell treatments

Cells, knocked-down cells, or plasmid-transfected cells were mock-infected or infected with RSV (MOI = 1) and simultaneously treated with U18666A (10 μM), MβCD (100 μM), orlistat (10 μM), CQ (5 μM), RAPA (200 nM), BafA1 (200 nM), lovastatin (25 μM), or water-soluble cholesterol (40 μM; Sigma-Aldrich, St. Louis, MO, USA) in normal medium or cholesterol-free medium for the indicated durations.

## Plasmids, siRNA, and transfection

The pGL3-SRE, pRT-TK, mCherry-OSBPL1A, mCherry-OSBPL1A (1-514aa), mCherry-OSBPL1A (1-514aa)YDAA, EGFP-VAPA, mCherry-RILP, TagBFP-RILP, mCherry-RILP (199-401aa), TagBFP-RILP (199-401aa), mCherry-DCTN1, EGFP-DCTN1, EGFP-Rab7A, EGFP-Rab7A Q67L, EGFP-

Rab7A T22N, mCherry-PLEKHM1, TagBFP-PLEKHM1, EGFP-VPS39, EGFP-VPS41, and mRFP-GFP-LC3B plasmids were constructed from MiaoLingBio (Hubei, China). siRNAs (si-ORP1L and si-LDLR) were acquired from RiboBio Co., Ltd. (Guangzhou, China). The siRNA sequences are listed in Supplementary Table 1.

According to the manufacturer's instructions, siRNAs or plasmids were transfected into HEp-2, 16HBE, HBECs, or HEK293T cells using Lipofectamine™ 6000 transfection reagent (Beyotime Biotechnology). The siRNAs or plasmids were diluted in Opti-MEM and then mixed with Opti-MEM containing Lipofectamine 6000. After 20-min incubation, the mixture was added to the cells. At 6 h post-transfection, the mixture was replaced with fresh medium. The related experiments were carried out 24 h or 48 h after transfection.

## Immunofluorescence assay

The monolayer was fixed in 4% paraformaldehyde for 15 min, exposed to 0.1% Triton X-100 for 15 min, and incubated in PBS containing 5% bovine serum albumin (BSA) for 1 h. The monolayer was then stained with the indicated antibodies overnight at 4 °C, followed by treatment with fluorescently conjugated secondary antibodies for 2 h at room temperature. Nuclear DNA was labeled with DAPI. Finally, the fluorescence intensity was measured, and the confocal or STED images were acquired using an LSM800 confocal microscope with Airyscan (Carl Zeiss AG, Oberkochen, Germany), Multimodality Structured Illumination Microscopy X (NanoInsights-Tech Co., Ltd., Guangzhou, China), or Abberior STEDYCON (Abberior Instruments GmbH, Göttingen, Germany), respectively. Imaging and image processing were done with ImageJ (National Institutes of Health, Bethesda, MD, USA) and Microscopy Image Analysis Software 10.1 (Imaris 10.1, Raybio Medical Technology, Guangzhou, China).

## RT-PCR assay

Total RNA was extracted from cells or mouse lung tissues and reverse-transcribed into cDNA using a PrimeScript™ RT reagent kit (TaKaRa Bio Inc., Kusatsu, Japan). PCR amplification was performed with TB Green Premix Ex Taq (TaKaRa Bio Inc.) on a fluorescence-based quantitative PCR system (Roche, Pleasanton, CA, USA). The primer pairs are listed in Supplementary Table 2.

## Lysosomal extraction assay

Lysosomes were extracted using a lysosomal extraction kit (X−Y Biotechnology, Shanghai, China) according to the manufacturer's instructions. Briefly, HEp-2 or HBEC cells were mock-infected or infected with RSV (MOI = 1) for 24 h. Then, the cells were washed twice with PBS and collected via centrifugation at 400×g at 4 °C for 5 min. Next, the cells were exposed to lysosomal extraction reagent A and incubated at 4 °C for 10 min. The cell suspension was homogenized 40 times with a homogenizer, and the supernatant was collected via centrifugation at 3000×g and 5000×g at 4 °C for 10 min. Afterward, the supernatant was centrifuged at 16,000×g at 4 °C for 20 min. The pellet was resuspended with reagent B and centrifuged again at 16,000×g for 20 min. The pellet was stored in reagent C and analyzed using western blotting.

## Western blotting assay

The monolayer was lysed using a cell lysis mixture (100× PMSF, 20× protease inhibitor, 20× protease-phosphatase inhibitor, and cell lysis solution) on ice for 30 min and then centrifuged at 14,000×g at 4 °C for 15 min. The total protein in each group was collected, and its concentration was determined using a BCA protein assay kit (Thermo Fisher Scientific). After separation using SDS-PAGE, the proteins were transferred onto polyvinylidene fluoride membranes. The membranes were blocked using TBST containing 5% BSA and stained with the indicated primary antibodies at 4 °C overnight. After washing thrice with TBST, the membranes were incubated

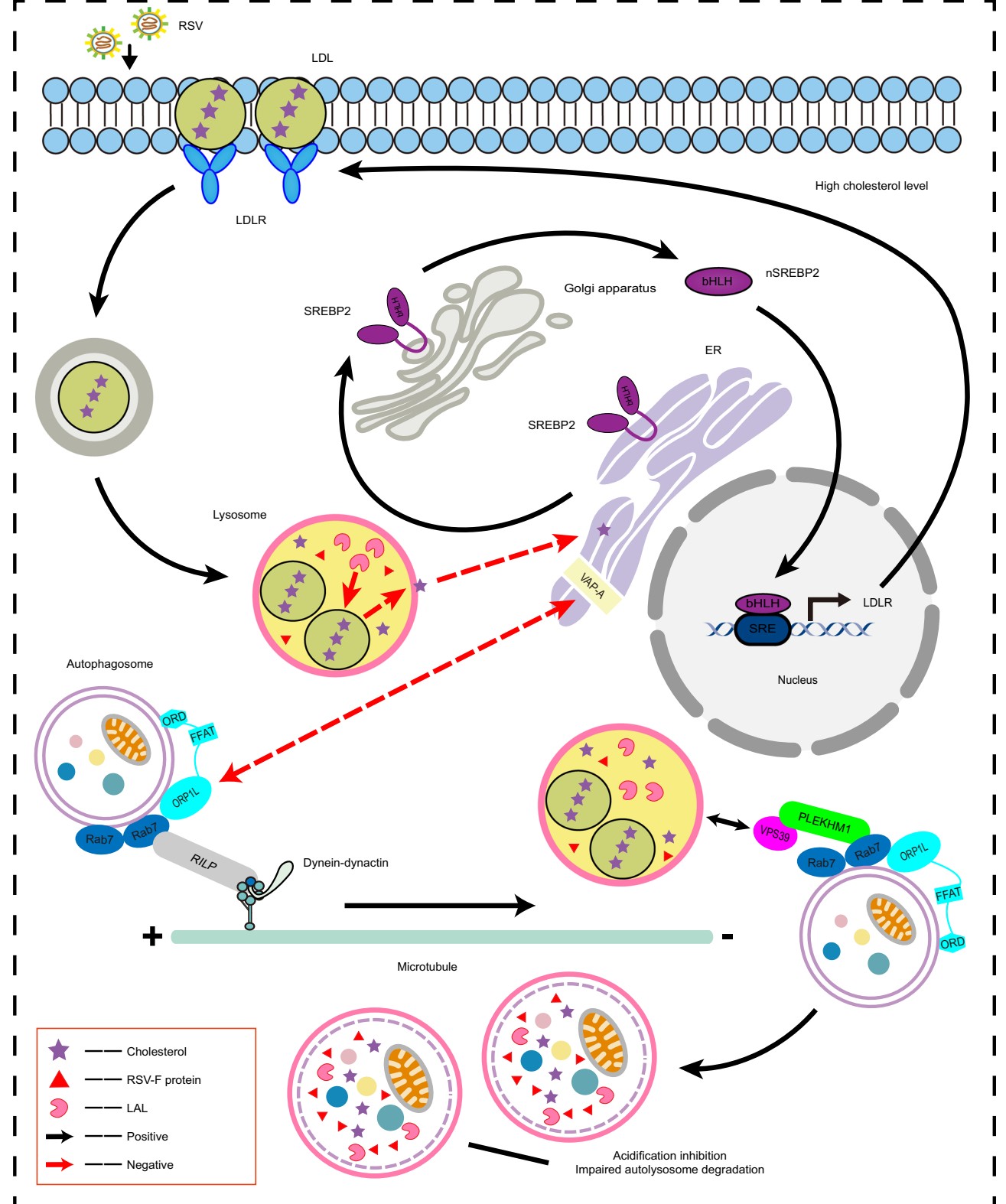

**Fig. 9 | Diagram depicting RSV-driven cholesterol reprogramming in lysosomes to facilitate viral replication.** RSV reprograms cholesterol metabolism in infected cells. In brief, RSV blocks cholesterol transport from lysosomes to the ER by reducing LAL activity, activates the SREBP2–LDLR axis, and promotes the uptake and accumulation of exogenous cholesterol in lysosomes in infected cells. The high cholesterol level in infected cells triggers the minus-end transport of autophagosomes and autolysosome formation by altering ORP1L function. On the other hand, cholesterol-rich lysosomes show acidification inhibition and dysfunction and impair autolysosome degradation and autophagic flux, which create sites for RSV-F protein storage. Inhibition of LDLR effectively reverses RSV-induced increase in cholesterol content, thereby blocking RSV infection.

with secondary antibodies for 2 h at room temperature. Finally, the protein bands were visualized under enhanced chemiluminescence using an Amersham Imager 600 (General Electric Co., Boston, MA, USA) and quantified using ImageJ (National Institutes of Health).

## Cholesterol estimation assay

The monolayer was lysed as described above, and the total cholesterol content was determined using an Amplex™ Red Cholesterol Assay Kit (Thermo Fisher Scientific, Waltham, MA, USA) according to the manufacturer's instructions. Briefly, cholesterol-containing samples (45 μL 1× reaction buffer + 5 μL sample) and standard solution were added to a 96-well plate. The reaction was initiated by adding 50 μL of the enzyme working solution to each well and continued at 37 °C for 30 min. The fluorescence signal was then measured in a microplate reader (550 nm excitation, 590 nm emission) (Thermo Fisher Scientific).

## LAL assay

LAL activity was determined using a LysoLive™ Lysosomal Acid Lipase Assay Kit (Abcam) according to the manufacturer's instructions. Briefly, HEp-2 cells were mock-infected or infected with RSV (MOI = 1) for 24 h. Orlistat (10 μM) was used as a control. The monolayer was then treated with LysoLive™ LipaGreen™ (1 μL per well) and incubated at 37 °C for 4 h. After resuspending cells with 500 μL 1× flow holding and sorting buffer, the fluorescence signal of LipaGreen™ was detected on a flow cytometer using the FITC channel (BD Biosciences, San Jose, CA, USA).

## Dual-luciferase reporter assay

HEK293T cells transiently expressing pGL3-SRE-LUC and pRL-TK were mock-infected or infected with RSV (MOI = 1) for 6 h, 12 h, 24 h, or 48 h. Firefly and Renilla luciferase activities were measured via a dual-luciferase reporter assay system (Beyotime Biotechnology, Shanghai, China).

## Dil-LDL uptake assay

HEp-2 cells were mock-infected or infected with RSV (MOI = 1) for 0 h, 1 h, 2 h, 6 h, 12 h, 24 h, or 48 h. U18666A (10 μM) was used as a control. The monolayer was then treated with Dil-LDL (30 μg/mL; Solarbio Science & Technology Co.,Ltd., Beijing, China) and incubated at 37 °C for 30 min. After washing twice with PBS, the cells were collected, and the fluorescence signal of Dil-LDL was detected on a flow cytometer using the FITC channel (BD Biosciences).

## mRFP-GFP-LC3B assay

HEK293T cells transiently expressing mRFP-GFP-LC3B were mock-infected or infected with RSV (MOI = 1) in the presence or absence of cholesterol or MβCD (100 μM) for 24 h. U18666A (10 μM), CQ (5 μM), and RAPA (200 nM) were used as controls. The monolayers were then fixed in 4% paraformaldehyde for 15 min. After washing the cells twice with PBS, the fluorescence intensity of mRFP-GFP-LC3B was detected, and images were obtained using a confocal microscope with Airyscan (Carl Zeiss AG).

## LysoSensor staining assay

LysoSensor dye (Yeasen, Shanghai, China) stains cellular acidic compartments and indicates the acidity of lysosomes. Briefly, HEp-2, 16HBE, or HBECs cells were mock-infected or infected with RSV (MOI = 1) in the presence or absence of cholesterol or MβCD (100 μM) for 24 h. CQ (5 μM) was used as a control. The monolayer was then treated with LysoTracker (10 μM) and LysoSensor (10 μM) and incubated at 37 °C for 30 min. The monolayers were fixed in 4% paraformaldehyde for 15 min. After washing the cells twice with PBS, the fluorescence intensity of LysoSensor was detected, and images

were obtained using a confocal microscope with Airyscan (Carl Zeiss AG).

## Lysosomal intracellular activity assay

Lysosomal activity was determined using a Lysosomal Intracellular Activity Assay Kit (Abcam) according to the manufacturer's instructions. Briefly, HEp-2 cells were mock-infected or infected with RSV (MOI = 1) in the presence or absence of cholesterol or MβCD (100 μM) for 24 h. U18666A (10 μM), CQ (5 μM), and BafA1 (200 nM) were used as controls. The monolayers were then treated with self-quenched substrate (15 μL per well) and incubated at 37 °C for 1 h. After washing the cells twice with 1 mL ice-cold 1× assay buffer, the fluorescence signal of the self-quenched substrate was detected on a flow cytometer using the FITC channel (BD Biosciences).

## TEM

The effect of RSV on autolysosomes was determined using TEM. Briefly, HEp-2, 16HBE, or HBEC cells were mock-infected or infected with RSV (MOI = 1) for 24 h. The cells were then collected, washed thrice with PBS, and fixed with 2.5% glutaraldehyde at 4 °C for 30 min. The pellets were dehydrated using an acetone series and embedded in epoxy resin. Images of the autolysosome were obtained using a Hitachi H-7500 (HITACHI, Tokyo, Japan).

## Cell viability assay

The cytotoxicity of si-LDLR was determined using a cell counting kit-8 (CCK-8; Yeasen Biotechnology, Shanghai, China) according to the manufacturer's instructions. Briefly, confluent HEp-2, 16HBE, or HBECs cells in a 96-well plate were transfected with si-LDLR (50 nM) or si-NC (50 nM) and incubated at 37 °C for 48 h. Next, 10 μL of CCK-8 was added to each well. After 30 min of incubation at 37 °C, the absorbance values at 450 nm were detected using a microplate reader (Thermo Fisher Scientific). Cell viability was expressed as a percentage of non-transfected controls.

## Animals and experimental protocol

Three-week-old male C57BL/6 mice (WT) or LDLR$^{-/-}$ C57BL/6 mice (Genetic background: 000664 C57BL/6 J) were purchased from Cavens (Jiangsu, China) and maintained in a pathogen-free room (a 12 h light/dark cycle from 6:00 am to 6:00 pm and temperature of 22 ± 2 °C with 40–70% humidity) with free access to water and food. Mice were randomly divided into three groups (n = 6 for each group): (a) normal control (without RSV infection), (b) RSV control, and (c) RSV + LDLR$^{-/-}$ group. The mice were anesthetized and intranasally infected with RSV. On day 4 after RSV infection, the mice were weighed and euthanized, and the lung tissues were aseptically excised and weighed. The lung index was expressed as lung weight/body weight. The lung tissues were then used for plaque assays, RT-PCR, immunofluorescence staining, and histopathological examination. Blood samples were collected and used to measure the levels of inflammatory cells. All mice were handled in strict adherence to the Guidelines for Laboratory Animal Use and Care of the Chinese Center for Disease Control and Prevention (CDC) and the Rules for Medical Laboratory Animals of the Ministry of Health, China. The animal study protocol was approved by the Ethics Committee of Jinan University and the National Institute for Communicable Disease Control and Prevention. The institution reference ID for animal ethics approval is 20220314-14.

## Statistics and reproducibility

Each experiment was repeated independently at least three times with similar results. Data are expressed as means ± SD using the GraphPad Prism v.9.4 software (GraphPad Software, La Jolla, CA, USA). One-way ANOVA followed by Tukey's test or Student's t-test (two-sided) was used for statistical comparisons. P < 0.05 was considered to indicate statistical significance.

**Reporting summary**

Further information on research design is available in the Nature Portfolio Reporting Summary linked to this article.

## Data availability

Source data are provided in this paper.

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

## Acknowledgements

This work was supported by the National Natural Science Foundation of China (82073895/82274180: L.M., 82204464: C.L., and 82273979: L.Z.), Basic and Applied Basic Research Foundation of Guangdong (2021A1515220126: L.Z.), and National Key Research and Development Program of China (2021YFC2300400: Z.H.).

## Author contributions

Zhong Liu, Manmei Li, and Hong Zhang conceived and designed the study; Lifeng Chen, Jingjing Zhang, Weibin Xu, Jiayi Chen, Yujun Tang, and Si Xiong performed the biological experiments and collected the data; Zhong Liu and Lifeng Chen wrote the manuscript; Zhong Liu, Manmei Li, Hong Zhang, and Yaolan Li revised the work for intellectual content and context. All the authors read and approved the manuscript.

## Competing interests

The authors declare no competing interests.
