## [Peer Review File · Nature Communications]

Cholesterol-rich lysosomes induced by respiratory syncytial virus promote viral replication by blocking autophagy fluxREVIEWER COMMENTS

Reviewer #1 (Remarks to the Author):

The authors have submitted a manuscript that describes the mechanism by which RSV is able to inhibit proteolytic degradative roles of lysosomes while activating autophagy that is required for optimal viral replication. How RSV is able to activate autophagy though evade the degradative effects of the autolysosome is entirely unknown. This has been a mystery since the identification of autophagy as a mechanism required to prevent apoptosis of infected cells (PMID: 29386287, PMID: 34434268) and for optimal viral replication (PMID: 35038909, 34434268, 29386287) about 5 – 6 years ago.

The authors start their study of autophagy in Hep2 cells. In the Hep2 cells the authors observe RSV colocalization with LAMP1 proteins in human cells as we do also. They then go on to demonstrate that LDLR receptor KO leads to significantly lower RSV infection in C57 mice. The in vivo experiments in the LDLR mice support the central role of LDLR recruitment of cholesterol to mediate the effect. Note: The authors use, almost exclusively, Hep2 cells which were contaminated with HeLa cells as indicated on the ATCC website and so these are now thought to be HeLa cell in origin. The authors have also repeated some of their experiments in immortalized 293T cells.

They do an interesting experiment where they rescue RSV infection in cells silenced for LDLR expression using water soluble cholesterol. Another example of their scientific rigour is the use of inhibitor of NPC1 a cholesterol transporter as a reference control that phenocopies the transport of cholesterol during RSV infection.

The major problems that I have with the manuscript are the lack of primary cells to support the findings, image sampling of the confocal images, over-use of image contrast, and rigour. That is with regard to properly citing the foundation science discussed throughout the manuscript. I also feel that the authors try to cover too much science in this one paper. It could certainly do with some trimming of experimental data to focus the message of how RSV prevents autolysosome formation and degradative action.

All in all, This paper outlines an interesting discovery that is likely applicable to other viruses that activate autophagy. However I feel that it needs some further experimentation, data analysis and significant rewriting to support the central findings.

The major issues with the manuscript are described in more detail below:

1. There is no use of primary cells within the manuscript. Currently the study consists of Hep2 that were contaminated with HeLa cells in the 1990s and so they are cervical carcinoma cells. The other cell type are HEK 293T embryonic kidney cells. Primary cells and a bronchial epithelial cell type are needed throughout the manuscript to support some of the mechanistic findings. The animal experiments support the results but the LDLR KO could be doing a myriad of different things during RSV infection. Bronchial epithelial cells are not easily transfected but they can be treated with the inhibitors to support the major findings in the manuscript.

2. For the majority of the manuscript the authors use image analysis of immuno- fluorescence confocal images. This is encouraged to enhance a manuscript. However if one is to use this as the primary way to come to scientific conclusions much scrutiny is required. The authors need to describe the optical resolution and the voxel size of the images. Many criteria of biological imaging have been published but one that is more common is the Nyquist criteria. To put this simply, the voxel size must be smaller than the optical resolution. Otherwise the authors run the risk of undersampling single cells which runs the risk of false positives while conducting colocalization analysis. In some of the images , particularly those in figure 1a, the authors need to zoom in further to fulfil this criteria. Consistency is crucial as well, In the mock they image only one or two cells but in the other panels they image several. The RSV-N images in fig 1a look undersampled and so it is possible this is a false positive. Again, I feel that the resolution is too low in some of these images and need to be repeated at higher resolution with a Nyquist calculation to estimate the proper pixel and voxel sampling size.

3. An example of desirable image appearance/contrast is that shown in Figure 3d. The green channel images in most of the panels show what appears to be over-use of contrast adjustment. Contrast adjustment is completely fine if done to the same degree across all images and panels and to increase ability to discern structures but it looks like inappropriate image manipulation where the researcher is trying to hide something. Therefore a lot of these images need to be reprocessed with less contrast adjustment and the authors must read and adhere to the journals guidance on biological imaging.

The images in question in this regard are in Figure 1 as mentioned above, figure 2, figure 3k, figure 4b, figure 5.

4. There is a paucity of references for comments on the foundation science made in the manuscript. For example references are needed for comments like, "Second, NPC1 and NPC2 also serve as key factors in regulating cholesterol transport to the lysosomal limiting membrane." The authors must reference those who discovered this first and all other comments like it throughout the manuscript.

Minor comments:

5. The figures must be labelled with the figure number on the figure itself for ease of review. That is, 'Figure 1,' 'Figure 2,' 'Figure 3 and so on.

6. In figure 8 there is no need for the red arrows.

7. It appears as though the data in fig 8 a, c are not consistent with the difference in the images of the infection in fig 8d. It also looks as though there is unacceptable image contrast. Unacceptable is where image contrast has been increased past the point where data starts to be hidden. Though I am sure it is not to mislead but merely overzealous/naive. Nevertheless this should be corrected.

8. The authors may consider renaming the NPC1 inhibitor in the manuscript to 'NPC1i' or 'NPC1 inh' throughout. This way the identity of the inhibitor is apparent without having to scroll through the manuscript trying to find out what "U18666A" means.

Reviewer #2 (Remarks to the Author):

This is an interested manuscript describing the lysosomal biology of RSV virus and at the same time showing a potential way to apply this knowledge in manipulation of host cells to prevent efficient infection.

The authors start by showing how RSV affects the cholesterol content in lysosomes and then make the argument that the consequence is altered transport and fusion with phagosomes, increase of gene expression for proteins related to cholesterol uptake (like the LDL receptor) and attenuation of the autophagic flux. As a consequence, proteins from the infecting RSV virus are more stable (and the authors showed that RSV transcription as well as cellular uptake are not affected). Manipulation of the LDLreceptor then controls infection in tissue culture and in vivo, which illustrates the pivotal role of cholesterol in RSV infection.

This is a very interesting study. My main problem is that it remains confusing as to the way lysosomal cholesterol -or better- authophagosomal/lysosomal RSV protein storage has something to do with the infection mechanism as these viruses ultimately have to enter the cytosol/nucleus for propagation. This point should be made more clear.

Some specific points.

General: Magnification of fluorescent images are different, based on the size of the nucleus. Please correct

All figures: what is the colocalization score really doing. And are the three balls in the figure, the score for 3 cells or three independent experiments? This information should be provided.

Figure 1A,B why is the distribution of the three RSV proteins evaluated different? And what will be the implications?

Figure 6g. The EM is nice but the results difficult to interpret without high magnification zoom-ins and without quantification. How do the authors identify autolysosomes on the basis of morphology?

Reviewer #1 (Remarks to the Author):

The authors have submitted a manuscript that describes the mechanism by which RSV is able to inhibit proteolytic degradative roles of lysosomes while activating autophagy that is required for optimal viral replication. How RSV is able to activate autophagy though evade the degradative effects of the autolysosome is entirely unknown. This has been a mystery since the identification of autophagy as a mechanism required to prevent apoptosis of infected cells (PMID: 29386287, PMID: 34434268) and for optimal viral replication (PMID: 35038909, 34434268, 29386287) about 5 – 6 years ago.

The authors start their study of autophagy in Hep2 cells. In the Hep2 cells the authors observe RSV colocalization with LAMP1 proteins in human cells as we do also. They then go on to demonstrate that LDLR receptor KO leads to significantly lower RSV infection in C57 mice. The in vivo experiments in the LDLR mice support the central role of LDLR recruitment of cholesterol to mediate the effect. Note: The authors use, almost exclusively, Hep2 cells which were contaminated with HeLa cells as indicated on the ATCC website and so these are now thought to be HeLa cell in origin. The authors have also repeated some of their experiments in immortalized 293T cells.

They do an interesting experiment where they rescue RSV infection in cells silenced for LDLR expression using water soluble cholesterol. Another example of their

scientific rigour is the use of inhibitor of NPC1 a cholesterol transporter as a reference control that phenocopies the transport of cholesterol during RSV infection.

The major problems that I have with the manuscript are the lack of primary cells to support the findings, image sampling of the confocal images, over-use of image contrast, and rigour. That is with regard to properly citing the foundation science discussed throughout the manuscript. I also feel that the authors try to cover too much science in this one paper. It could certainly do with some trimming of experimental data to focus the message of how RSV prevents autolysosome formation and degradative action.

All in all, This paper outlines an interesting discovery that is likely applicable to other viruses that activate autophagy. However I feel that it needs some further experimentation, data analysis and significant rewriting to support the central findings.

The major issues with the manuscript are described in more detail below:

1. There is no use of primary cells within the manuscript. Currently the study consists of Hep2 that were contaminated with HeLa cells in the 1990s and so they are cervical carcinoma cells. The other cell type are HEK 293T embryonic kidney cells. Primary cells and a bronchial epithelial cell type are needed throughout the manuscript to support some of the mechanistic findings. The

animal experiments support the results but the LDLR KO could be doing a myriad of different things during RSV infection. Bronchial epithelial cells are not easily transfected but they can be treated with the inhibitors to support the major findings in the manuscript.

Response: Thanks for the good suggestion, normal primary human bronchial epithelial cells (HBECs) and human bronchial epithelial cells (16HBE) had been used to support some of the mechanistic findings (As shown in Fig. 1a-f and h, Fig. 3k-m, Fig. 5l-o, Fig. 6b-g and j-o, Fig. 7a-f and h-l, and Fig. S2a-i).

2. For the majority of the manuscript the authors use image analysis of immunofluorescence confocal images. This is encouraged to enhance a manuscript. However if one is to use this as the primary way to come to scientific conclusions much scrutiny is required. The authors need to describe the optical resolution and the voxel size of the images. Many criteria of biological imaging have been published but one that is more common is the Nyquist criteria. To put this simply, the voxel size must be smaller than the optical resolution. Otherwise the authors run the risk of undersampling single cells which runs the risk of false positives while conducting colocalization analysis. In some of the images, particularly those in figure 1a, the authors need to zoom in further to fulfil this criteria. Consistency is crucial as well, In the mock they image only one or two cells but in the other panels they image several. The RSV-N images in fig 1a look undersampled and so it is possible this is a false positive. Again, I feel that the

resolution is too low in some of these images and need to be repeated at higher resolution with a Nyquist calculation to estimate the proper pixel and voxel sampling size.

Response: Thanks for the good suggestion, the majority of immunofluorescence assays had been repeated using a LSM800 confocal microscope with Airyscan, Multimodality Structured Illumination Microscopy X, or Abberior STEDYCON (As shown in **Fig. 1a-g, Fig. 3d and k-m, Fig. 4, Fig. 5, and Fig. 6a-g**). The images have already met the Nyquist criteria and have higher resolution. The image parameters had been described in the figure legends. Some of the images had been zoomed (As shown in **Fig. 1a, c, e, and g, Fig. 4b and e, and Fig. 5**). Moreover, the number of cells in each group had been consistent. The RSV-N images obtained using anti-respiratory syncytial virus nucleoprotein antibodies had been further verified and no problems were found.

3. An example of desirable image appearance/contrast is that shown in Figure 3d. The green channel images in most of the panels show what appears to be over-use of contrast adjustment. Contrast adjustment is completely fine if done to the same degree across all images and panels and to increase ability to discern structures but it looks like inappropriate image manipulation where the researcher is trying to hide something. Therefore a lot of these images need to be reprocessed with less contrast adjustment and the authors must read and adhere to the journals guidance on biological imaging.

The images in question in this regard are in Figure 1 as mentioned above, figure 2, figure 3k, figure 4b, figure 5.

Response: The majority of immunofluorescence assays had been repeated, and processing (changing brightness and contrast) is applied equally across the entire image or is applied equally to controls. In fact, some of the immunofluorescence assays in the manuscript had been verified by other methods. **(1)** The lysosomal extraction kit had been used to extract lysosomes from RSV-infected cells (HEp-2 and HBECs), and the results of **Fig. 1h** showed that the F protein is enriched in lysosomes, which further verifies the results of **Fig. 1a, c, e**. **(2)** **Fig. 3k-m** had been repeated using Abberior STEDYCON, and the results showed that Dil-LDL (exogenous cholesterol) is trapped in the lysosomal lumen, which further verifies the results of **Fig. 1a, c, e** and **Fig. 2a**. **(3)** The results of **Fig. 3d** had also been further verified by **Fig. 3e**.

4. There is a paucity of references for comments on the foundation science made in the manuscript. For example references are needed for comments like, “Second, NPC1 and NPC2 also serve as key factors in regulating cholesterol transport to the lysosomal limiting membrane.” The authors must reference those who discovered this first and all other comments like it throughout the manuscript.

Response: Thanks for the good suggestion, the relevant references had been inserted into the manuscript (As shown in **line 47, 53, 58, 107, 118, 150, 152, 157, 168, 205, 208, 216, 285, 292, 295, 311, 350, 430, and 433**).

Minor comments:

5. The figures must be labelled with the figure number on the figure itself for ease of review. That is, 'Figure 1,' 'Figure 2,' 'Figure 3 and so on.

Response: Each figure picture had been named.

6. In figure 8 there is no need for the red arrows.

Response: The red arrows in Fig. 8d had been deleted.

7. It appears as though the data in fig 8 a, c are not consistent with the difference in the images of the infection in fig 8d. It also looks as though there is unacceptable image contrast. Unacceptable is where image contrast has been increased past the point where data starts to be hidden. Though I am sure it is not to mislead but merely overzealous/naive. Nevertheless this should be corrected.

Response: The results of Fig. 8a were obtained using plaque assay. In this assay, the viral titer close to 2 indicates that > 90 % of the virus particles have been cleared. Therefore, the results of Fig. 8a are consistent with those of Fig. 8c, d. Moreover, the contrast adjustment in Fig. 8d is to remove the background of the control group, and processing (changing brightness and contrast) is applied equally across the entire image and is applied equally to controls. The image contrast had been corrected, and the unprocessed data had been shown in Fig. 8d.

8. The authors may consider renaming the NPC1 inhibitor in the manuscript to ‘NPC1i’ or ‘NPC1 inh’ throughout. This way the identity of the inhibitor is apparent without having to scroll through the manuscript trying to find out what “U18666A” means.

Response: U18666A is the official name of a NPC1 inhibitor and is widely used in other literatures. “U18666A means a NPC1 inhibitor” had been shown in **lines 116-118.**

Reviewer #2 (Remarks to the Author):

This is an interested manuscript describing the lysosomal biology of RSV virus and at the same time showing a potential way to apply this knowledge in manipulation of host cells to prevent efficient infection.

The authors start by showing how RSV affects the cholesterol content in lysosomes and then make the argument that the consequence is altered transport and fusion with phagosomes, increase of gene expression for proteins related to cholesterol uptake (like the LDL receptor) and attenuation of the autophagic flux. As a consequence, proteins from the infecting RSV virus are more stable (and the authors showed that RSV transcription as well as cellular uptake are not affected). Manipulation of the LDLreceptor then controls infection in tissue culture and in vivo, which illustrates the pivotal role of cholesterol in RSV infection.

The issues with the manuscript are described in more detail below:

1. This is a very interesting study. My main problem is that it remains confusing as to the way lysosomal cholesterol -or better- autophagosomal/lysosomal RSV protein storage has something to do with the infection mechanism as these viruses ultimately have to enter the cytosol/nucleus for propagation. This point should be made more clear.

Response: Thanks for the good suggestion, normal primary human bronchial epithelial cells (HBECs) and human bronchial epithelial cells (16HBE) had been used to support some of the mechanistic findings (As shown in Fig. 1a-f and h, Fig. 3k-m, Fig. 5l-o, Fig. 6b-g and j-o, Fig. 7a-f and h-l, and Fig. S2a-i). The results in this study showed that RSV-F proteins but not the other viral proteins (such as RSV-G and RSV-N) are stored in cholesterol-rich lysosomes in normal primary human bronchial epithelial cells (HBECs) and human bronchial epithelial cells (16HBE). This is the innovation of this manuscript, which reveals for the first time that RSV-F protein is stored in cholesterol-rich lysosomes of infected cells. In fact, the fusion protein (F) is a glycoprotein located on the surface of the virus and is essential for RSV replication. First, the F protein facilitates fusion of the viral membrane with the host-cell membrane, allowing the viral nucleic acid into the cytoplasm. Second, expression of RSV-F protein on the surface of infected cells can also trigger fusion with neighboring cells, which promotes the spread of progeny viruses between cells. In this study, we found that RSV promotes the storage and accumulation of RSV-F protein by inducing

the formation of cholesterol-rich lysosomes and inhibiting the degradation of autophagolysosomes, respectively. The amount of RSV-F protein in infected cells decreased significantly after cholesterol depletion or knockdown of LDLR. The lack of RSV-F protein means that the entry of progeny viruses into cells and their spread between cells will be hindered, ultimately affecting viral replication. Interestingly, supplementation of exogenous cholesterol restores RSV-F protein content in LDLR-knockdown cells. Therefore, RSV-mediated lysosomal cholesterol metabolism is mainly to protect RSV-F protein to exert its function. The related discussion had been added to the manuscript (As shown in lines 428-444).

2. General: Magnification of fluorescent images are different, based on the size of the nucleus. Please correct

Response: The majority of immuno-fluorescence assays had been repeated, and the magnification of the images is consistent.

3. All figures: what is the colocalization score really doing. And are the three balls in the figure, the score for 3 cells or three independent experiments? This information should be provided.

Response: The colocalization score means whether the two or three proteins are in the same location of the cell or whether they are in contact. The majority of immuno-fluorescence assays had been repeated, and the balls now represent the score for 15 or 20 images (2-4 cells per image). Data are one representative of three

independent experiments. The information had been provided in the figure legends.

4. Figure 1A,B why is the distribution of the three RSV proteins evaluated different? And what will be the implications?

Response: The results of Fig. 1a-f (Fig. 1a, b had been adjusted to Fig. 1a-f) had been repeated in normal primary human bronchial epithelial cells (HBECs) and human bronchial epithelial cells (16HBE) (As shown in Fig. 1a-f). The results showed that RSV-F proteins but not the other viral proteins (such as RSV-G and RSV-N) are stored in lysosomes. Moreover, the localization of these three proteins in cells is also different.

5. Figure 6g. The EM is nice but the results difficult to interpret without high magnification zoom-ins and without quantification. How do the authors identify autolysosomes on the basis of morphology?

Response: The EM assay had been repeated, and the results of Fig. 6j-o (Fig. 6g had been adjusted to Fig. 6j-o) had been highly magnified and quantified. Autolysosomes typically have only one limiting membrane; frequently they contain electron dense cytoplasmic material and/or organelles (Autophagy. 2016;12(1):1-222. doi: 10.1080/15548627.2015.1100356.).

REVIEWERS' COMMENTS

Reviewer #1 (Remarks to the Author):

The authors have done a lot, and an admirable amount, of work to improve the manuscript. The results and experiments are clear. However there are a lot of editorial issues with regard to the manuscript itself. This involves the need to remove some figures that are very much redundant or muddle the narrative of the paper. I feel that this is all par for the course of a paper this size to require significant editorial revision. The authors may wish to access an editorial service to streamline the paper and make it more succinct.

The following is a list of corrections but this is not exhaustive as there appears to be a lot of editorial work that needs to be done.

Line 80: RSV is not in the Paramyxovirus family. It is in the Pneumovirinae family. Please consult the appropriate literature and correct this error.

Line 34: "which enables RSV-F (fusion) protein hiding and accumulation." I suggest remove the word hiding and just state that this promotes accumulation of RSV-F.

Results section. Make sure to refer to the figures in a consistent manner in the results section. For example 1g is not explained in the results section. Figure references should appear earlier in lines 108 where the different cells are mentioned. Figure 1g may be removed. It does not appear to provide anything to the paper.

Figure 2A. I suggest removing this figure as it does not contribute much to the study other than it shows what figure 1 already does. It even confuses the points in the paper because it is difficult to determine the point the authors are trying to make by showing this data.

Their units are in inches whereas this should most likely be in μm because this is the scale of a cell, not inches. Furthermore, inches are an imperial unit of measure and not appropriate for a scientific paper.

Figure 3D. Most of the images can be removed from 3D to show the movement of the protein into the nucleus during infection. The western blot in 3A is sufficient.

Reviewer #2 (Remarks to the Author):

The data are strongly improved and I do not have any further points related to this beautiful manuscript.

Reviewer #1 (Remarks to the Author):

The authors have done a lot, and an admirable amount, of work to improve the manuscript. The results and experiments are clear. However there are a lot of editorial issues with regard to the manuscript itself. This involves the need to remove some figures that are very much redundant or muddle the narrative of the paper. I feel that this is all par for the course of a paper this size to require significant editorial revision. The authors may wish to access an editorial service to streamline the paper and make it more succinct.

The following is a list of corrections but this is not exhaustive as there appears to be a lot of editorial work that needs to be done.

1. Line 80: RSV is not in the Paramyxovirus family. It is in the Pneumovirinae family. Please consult the appropriate literature and correct this error.

Response: RSV belongs to the *Pneumovirus* genus of the *Paramyxoviridae* family. Therefore, both statements are correct (Nat Chem Biol. 2016 Feb;12(2):87-93.; Nat Rev Microbiol. 2023 Nov;21(11):734-749.). “Human respiratory syncytial virus (RSV) is an enveloped, negative-sense RNA virus of the *Paramyxoviridae* family” had been modified to “Human respiratory syncytial virus (RSV) is an enveloped, negative-sense RNA virus that belongs to the *Pneumovirus* genus of the *Paramyxoviridae* family” (As shown in lines 79-80).

2. Line 34: “which enables RSV-F (fusion) protein hiding and accumulation.” I suggest remove the word hiding and just state that this promotes accumulation of RSV-F.

Response: The sentence “which enables RSV-F (fusion) protein hiding and accumulation” had been modified (As shown in lines 34-35, 130-131, 341-342, and 420).

3. Results section. Make sure to refer to the figures in a consistent manner in the results section. For example 1g is not explained in the results section. Figure references should appear earlier in lines 108 where the different cells are mentioned. Figure 1g may be removed. It does not appear to provide anything to the paper.

Response: The purpose of Fig. 1g (Fig. 1g had been adjusted to Supplementary Fig. 1) is to confirm that cholesterol fails to reach the ER. This is the main reason for the activation of SREBP2–LDLR axis after RSV infection of host cells. Fig. 1g had been converted to supplementary material (Supplementary Fig. 1).

4. Figure 2A. I suggest removing this figure as it does not contribute much to the study other than it shows what figure 1 already does. It even confuses the points in the paper because it is difficult to determine the point the authors are trying to make by showing this data.

Response: Thanks for the good suggestion, Fig. 2a had been removed.

5. Their units are in inches whereas this should most likely be in um because this is the scale of a cell, not inches. Furthermore, inches are an imperial unit of measure and not appropriate for a scientific paper.

Response: Thanks for the good suggestion, “inches” had been revised. (As shown in Fig. 6a, Supplementary Fig. 1, and Supplementary Fig. 4).

6. Figure 3D. Most of the images can be removed from 3D to show the movement of the protein into the nucleus during infection. The western blot in 3A is sufficient.

Response: Thanks for the good suggestion, Fig. 3d had been removed.

Reviewer #2 (Remarks to the Author):

The data are strongly improved and I do not have any further points related to this beautiful manuscript.

Response: Thank you.